# Profiling stress-triggered RNA condensation with photocatalytic proximity labeling

Ziqi Ren[1,4], Wei Tang[2,4], Luxin Peng [1] & Peng Zou [1,2,3] ✉

Stress granules (SGs) are highly dynamic cytoplasmic membrane-less organelles that assemble when cells are challenged by stress. RNA molecules are sorted into SGs where they play important roles in maintaining the structural stability of SGs and regulating gene expression. Herein, we apply a proximity-dependent RNA labeling method, CAP-seq, to comprehensively investigate the content of SG-proximal transcriptome in live mammalian cells. CAP-seq captures 457 and 822 RNAs in arsenite- and sorbitol-induced SGs in HEK293T cells, respectively, revealing that SG enrichment is positively correlated with RNA length and AU content, but negatively correlated with translation efficiency. The high spatial specificity of CAP-seq dataset is validated by single-molecule FISH imaging. We further apply CAP-seq to map dynamic changes in SG-proximal transcriptome along the time course of granule assembly and disassembly processes. Our data portray a model of AU-rich and translationally repressed SG nanostructure that are memorized long after the removal of stress.

Stress granules (SG) are highly dynamic cytoplasmic membrane-less organelles that function in various cellular processes including virial infection, stress response, etc. When cells are under stress that results in translational inhibition, microscopically visible SGs containing proteins and RNAs are assembled in a reversible manner[1,2]. SGs have been implicated in neurodegenerative diseases such as ALS and FTDP[3,4]. Mutations in SG proteins such as TDP-43 or TIA1 have been reported to reduce the dynamics of SG, resulting in resistance to clearance, which ultimately gives rise to cellular toxicity and neurodegeneration[5,6].

RNAs are key components of SGs. The SG-localization of RNAs have been associated with features including transcript lengths, nucleobase modifications like $N^6$-methyladenosine (m6A), and translation efficiency[7–9]. For example, m6A modification is found to be enriched in granules isolated from arsenite-stressed NIH-3T3 cells[8,10]. However, fluorescence microscopy analysis reveals a similar level of SG enrichment in RNA methylation-deficient mouse embryonic stem cells versus wild-type cells, leading to the conclusion that transcript lengths, rather than m6A levels, are correlated with SG enrichment[11].

It is generally believed that SG localized mRNAs are translationally repressed before entering SG and resume translation following SG disassembly[12–14]. Thus, sequestration of mRNA into SG may serve as a mechanism for translational control during stress response and cell survival. Yet, paradoxically, translation has also been observed within SGs[12,15], making the actual role of SGs and the targeting mechanism of RNAs more complicated. Thus, comprehensively resolving the RNAs composition of SGs under distinct stages of stress/recovery could shed light on their molecular function and the mechanism of RNA sorting into SGs.

Despite the importance of RNA localization in stress granules, methods for studying RNA localization have been limited. Imaging-based technique can directly visualize RNA in fixed and live cells to determine their localization and exchange rate with respect to SGs[15–17]. However, these methods require prior knowledge of RNA sequence and are thus less suitable for transcriptome-wide profiling of SG content. Biochemical purification of SG core component has been achieved in yeast and mammalian cells[18,19], unveiling the features of SG transcriptome. However, the purification procedure is prone to

[1]College of Chemistry and Molecular Engineering, Synthetic and Functional Biomolecules Center, Beijing National Laboratory for Molecular Sciences, Key Laboratory of Bioorganic Chemistry and Molecular Engineering of Ministry of Education, PKU-IDG/McGovern Institute for Brain Research, Peking University, Beijing 100871, China. [2]Academy for Advanced Interdisciplinary Studies, Peking-Tsinghua Center for Life Sciences, Peking University, Beijing 100871, China. [3]Chinese Institute for Brain Research (CIBR), Beijing 102206, China. [4]These authors contributed equally: Ziqi Ren, Wei Tang. ✉e-mail: zoupeng@pku.edu.cn

contamination and loss of weakly associated material, thus causing high false positive and high false negative rates, respectively. It is difficult to extend this approach to profile SG components during the disassembly stage.

Proximity labeling can label the targeting molecules directly in the region of interests which overcomes the shortcomings of fractionation-based methods. APEX2[20–22] and BioID[23] have been used in investigating the SG proteome[24]. However, BioID is incapable of labeling RNA, while APEX2[21,25] requires cytotoxic hydrogen peroxide at millimolar concentration, which is itself a cellular stress[26]. Recently, we have developed a photoactivatable proximity-dependent RNA labeling method, called CAP-seq[27]. Under ambient blue light illumination, genetically encoded photocatalyst miniSOG mediates the photo-oxidization of RNA, which could be covalently captured with an exogenously supplied amine probe. Due to the limited half-life and diffusion distance of singlet oxygen (0.6 μs and 70 nm in cells)[28], CAP-seq have been applied in profiling the subcellular transcriptome in the mitochondria and endoplasmic reticulum with high spatial resolution in live cells[27]. As the size of SGs is in the range of 100 nm–1 μm[3], a resolution on the scale of tens of nanometers is sufficient for mapping SG components. Notably, CAP-seq is capable of labeling RNAs that are indirectly interacting with the bait, which could complement direct capture methods such as chemical- or photo-crosslinking (e.g., CLIP, PAR-CLIP)[29,30].

Herein, we applied CAP-seq to systematically profile SG transcriptome in live cells (Fig. 1A). Our data reveal that the targeting efficiency of mRNA into stress granules is positively correlated with transcript length, AU content and m6A abundance. We also observed stress-specific SG targeting of mRNAs when cells are challenged by oxidative stress (e.g., sodium arsenite) or osmotic shock (e.g., sorbitol). The high spatial specificity of CAP-seq allows profiling of RNA components in pre-existed, nanoscale SG cores, which are characterized by low translational efficiencies and more m6A sites. Application of CAP-seq to the SG disassembly process unveils the temporally dynamic RNA localization to SGs, portraying a model of AU-rich and translationally repressed SG nanostructures that are memorized long after the removal of stress (3 h post-stress).

## Results

### CAP-seq reveals SG transcriptome in HEK293T cells

We started by creating a HEK293T cell line stably expressing the fusion between miniSOG and G3BP1 (Ras GTPase-activating protein-binding protein 1), a core component of SG[31]. The expression level of G3BP1-miniSOG was 65 ± 5% (mean ± s.d.) that of the endogenous G3BP1 level, as measured by western blot (Supplementary Fig. 1). In a negative control HEK293T cell line, non-fused (thus untargeted) miniSOG is expressed throughout the whole cell. To induce stress granule (SG) formation, cells were treated with 0.5 mM sodium arsenite for 60 min to mimic oxidative stress. As sodium arsenite-induced eIF2α phosphorylation mediates the inhibition of translation initiation[32], we, therefore, monitored eIF2α phosphorylation levels in arsenite-stressed G3BP1-miniSOG HEK293T cells[32]. Western blot analysis reveals that CAP-seq labeling elevates eIF2α phosphorylation by ~200% from the basal level (Supplementary Fig. 2). Immunofluorescence imaging confirmed the co-localization of G3BP1-miniSOG (57 ± 7 %, mean ± s.d., n = 30 cells) with the SG marker, G3BP2, whereas untargeted miniSOG remains dispersed throughout the cell (21 ± 3 %, n = 30 cells) (Supplementary Fig. 3).

For RNA labeling, our previous CAP-seq experiments showed that 10 mM PA was capable of permeating throughout the cytoplasm within 5 min[27]. Since the core–shell structure of SG[1,2,18] might hinder probe penetration, we pre-incubated cells with 10 mM PA for 5 min and subsequently illuminated the sample with 24 mW/cm² blue LED for 15 min (Supplementary Fig. 4). We measured the level of eIF2α phosphorylation as a marker for translation inhibition. Western blot

analysis reveals that CAP-seq labeling elevates eIF2α phosphorylation by ~70% from the basal level. Treating cells with blue LED illumination alone or incubation with 10 mM PA alone causes the eIF2α phosphorylation levels to increase by ~40% and ~75%, respectively (Supplementary Fig. 5). These changes are modest as compared to arsenite-induced stress (Supplementary Fig. 2).

Immunofluorescence imaging showed highly co-localized signal between biotinylation and the SG marker G3BP2, thus demonstrating the high spatial specific of labeling (Fig. 1B). In negative control samples omitting either PA or light illumination, no biotinylation was observed. For cells expressing untargeted miniSOG, biotinylation signal was diffusive in the cell. RNA was extracted from cells and reacted with biotin-conjugated azide via copper-assisted alkyne azide cycloaddition (CuAAC). We chose THPTA as the ligand for CuAAC because it offers higher reaction yield with RNA in the aqueous solution[33]. Biotinylated RNA was enriched with streptavidin-coated magnetic beads and analyzed by next-generation sequencing (Fig. 1A). To remove background arising from non-specific adsorption on the magnetic beads, samples omitting PA was set as negative control. In addition, as G3BP1 was only partially localized to SG under sodium arsenite stress[34] (Supplementary Fig. 3), we used HEK293T cells expressing untargeted miniSOG as another control to subtract background RNA labeling in the cytoplasm.

We performed replicated experiments and applied DESeq2[35] analysis comparing the transcript abundance between labeled samples and relevant controls: 1) post- versus pre-enrichment of RNA labeled with G3BP1-miniSOG (post-enrichment: RNAs eluted from streptavidin-coated beads; pre-enrichment: total RNAs before loading onto streptavidin-coated beads); 2) RNA labeled with G3BP1-miniSOG versus RNA from negative control omitting PA; 3) RNA labeled with G3BP1-miniSOG versus untargeted miniSOG (Fig. 1C, Supplementary Fig. 6, Supplementary Data 1A). By applying a cutoff of log2FoldChange (log2FC > 0.3) and $p_{adj}$ < 0.05, the above analysis yielded 2018, 2633, and 1249 transcripts, respectively. The SG-proximal transcriptome was defined as the overlap of enriched RNA from the above three datasets, yielding a final list of 457 transcripts, 455 of which are mRNAs (Fig. 1D). In addition, a total of 705 and 1062 transcripts were significantly depleted (log2FC <−0.3, $p_{adj}$ < 0.05) in post- vs. pre-enrichment and G3BP1 vs. untargeted labeling datasets, respectively. We thus defined SG-excluded transcriptome as the overlap of the above lists, which contained 224 transcripts (Supplementary Fig. 7). As a demonstration of the high spatial specificity of CAP-seq, known SG-localized transcripts, such as lncRNA NORAD, mRNA DYNC1H1, and mRNA KDM5B[12,19] were enriched in our SG dataset, whereas mitochondrial RNAs were significantly depleted.

We next performed single-molecule FISH imaging to verify the subcellular localization of transcripts in our CAP-seq dataset. We chose APLP2, GAS1, BMS1 and USP7 from SG-proximal dataset, CCNL2 and PCBP2 from SG-excluded dataset. To visualize SG, EGFP is C-terminally fused with G3BP1 and stably expressed in HEK293T cells. Following sodium arsenite treatment for 1 h, cells were fixed with paraformaldehyde and stained with fluorescently labeled oligonucleotides targeting the above transcripts. The SG localization was quantified for each transcript by taking the ratio of FISH signal overlapping with SG over the total FISH signal. As expected, all four transcripts in the SG-proximal dataset are highly co-localized with SG, with nearly two-thirds of the FISH signal overlapping with SG signal (APLP2, 69 ± 7%; GAS1, 68 ± 11%; BMS1, 70 ± 8%; USP7, 65 ± 8%; mean ± s.d.) (Fig. 1E, F). In sharp contrast, transcripts in SG-excluded datasets do not appear aggregated in the SG, with less than 12% of the FISH signal overlapping with SG (Fig. 1E, F). For example, while some of the PCBP2 mRNA FISH signal falls within the boundary of SG, the overall FISH signal appears randomly distributed across the cytoplasm, without a clear sign of aggregation and specific SG targeting (Fig. 1E). Taken together, the

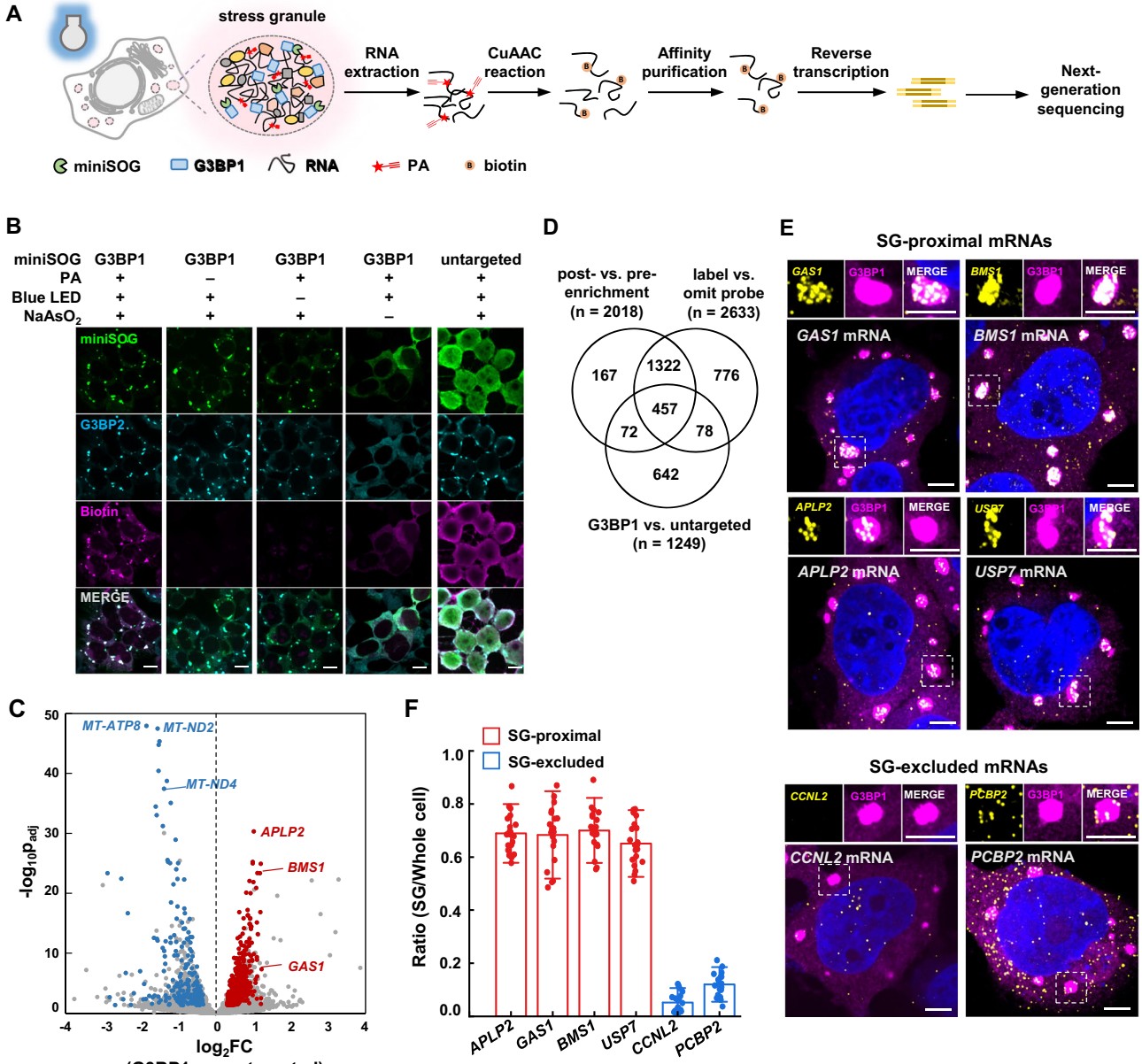

**Fig. 1 | Profiling SG transcriptome with CAP-seq. A** Schematic of SG CAP-seq workflow, with G3BP1 as the bait. **B** Immunofluorescence microscopy of HEK293T cells expressing G3BP1-miniSOG and untargeted miniSOG (green). Cells were treated with 0.5 mM sodium arsenite for 60 min and labeled by 10 mM PA under 15 min blue light illumination. Endogenous SG marker G3BP2 and biotinylated signal are shown in cyan and magenta, respectively. Unstressed controls were performed at a separate time. Images are representative examples from three independent experiments. Scale bars, 10 μm. **C** Volcano plot comparing the abundance of RNA captured with G3BP1-miniSOG vs. untargeted miniSOG in CAP-seq labeled HEK293T cells under arsenite stress. Red and blue dots represent SG-proximal and SG-excluded transcripts, respectively. **D** Venn diagram comparing RNAs enriched in: 1) post- vs. pre-enrichment of G3BP1-miniSOG CAP-seq; 2) G3BP1-miniSOG labeling vs. negative control omitting PA; 3) G3BP1-miniSOG vs. untargeted miniSOG. SG-proximal RNAs in arsenite-treated HEK293T are defined as the overlap of the above three datasets. **E** smFISH of SG-proximal (*GAS1*, *BMS1*, *APLP2*, *USP7*) and SG-excluded mRNAs (*CCNL2*, *PCBP2*). Scale bars, 5 μm. **F** Quantitation of the ratio of smFISH within SG vs. whole cell. The bars and lines represent mean values and standard deviations (SDs) respectively (*n* = 20 cells from three independent experiments).

above analysis demonstrated the high spatial specificity of our SG CAP-seq dataset.

## Comparing CAP-seq with purification-based methods in U-2 OS cells

The molecular inventory of SG has been investigated with conventional biochemical purification approaches[18,19]. For example, Parker et al. isolated sodium arsenite-induced SG from U-2 OS cells through differential centrifugation and antibody-based affinity purification[19]. To benchmark the coverage and specificity of our photocatalytic

proximity labeling method against purification-based approach in the same cell type, we repeated replicated CAP-seq in U-2 OS cells stably expressing G3BP1-miniSOG. Western blot analysis revealed that the expression level of G3BP1-miniSOG was approximately 63 ± 2% of the endogenous G3BP1 (Supplementary Fig. 8). SG formation was induced by the addition of 0.5 mM sodium arsenite for 60 min. Immunofluorescence imaging showed good co-localization between miniSOG and SG marker G3BP2, and 46 ± 3% (*n* = 30 cells) of the G3BP1-miniSOG fusion protein were localized to SGs (Supplementary Figure 9). In contrast, only 9.8 ± 0.4% of untargeted miniSOG protein (*n* = 30 cells)

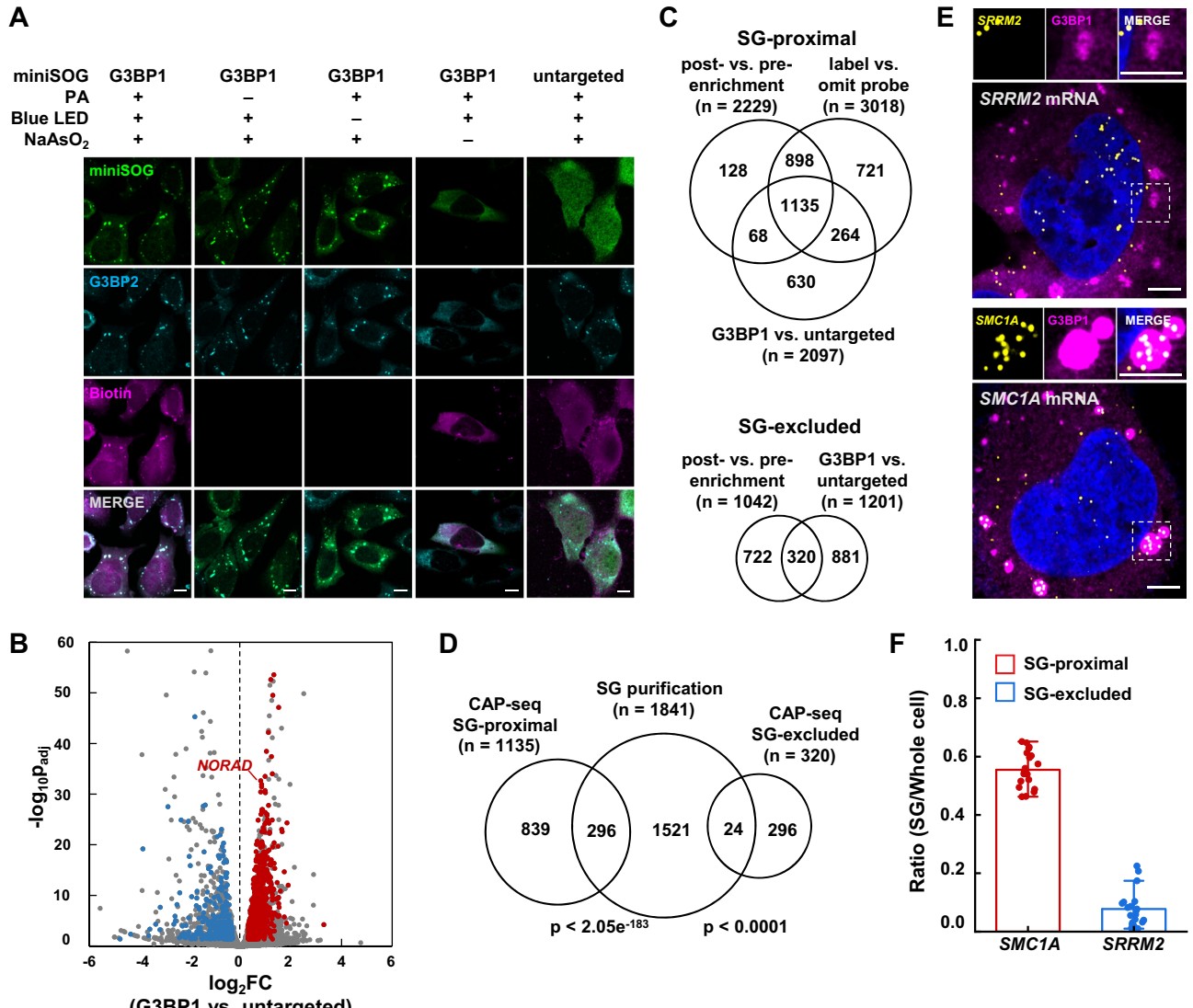

**Fig. 2 | Comparing CAP-seq with purification-based methods in U-2 OS cells.**
**A** Immunofluorescence microscopy of U-2 OS cells expressing G3BP1-miniSOG and untargeted miniSOG (green). Cells were treated with 0.5 mM sodium arsenite for 60 min and labeled by 10 mM PA under 15 min blue light illumination. Endogenous SG marker G3BP2 and biotinylated signal are shown in cyan and magenta, respectively. Unstressed controls were performed at a separate time. Images are representative examples from three independent experiments. Scale bars, 10 μm. **B** Volcano plot comparing the abundance of RNA captured with G3BP1-miniSOG vs. untargeted miniSOG in CAP-seq labeled U-2 OS cells under arsenite stress. Red and blue dots represent SG-proximal and SG-excluded transcripts, respectively. **C** Venn diagram comparing RNAs enriched in: (1) post- vs. pre-enrichment of G3BP1-miniSOG CAP-seq; (2) G3BP1-miniSOG labeling vs. negative control omitting PA; (3) G3BP1-miniSOG vs. untargeted miniSOG, SG-proximal RNAs in arsenite treated U-2 OS are defined as the overlap of the above three datasets; SG-excluded RNAs are defined as the overlap between Venn diagram comparing RNAs depleted in (1) and (3). **D** Venn diagram comparing CAP-seq SG-proximal and SG-excluded transcripts with SG transcriptome captured by purification-based method[19]. The p values were calculated by hypergeometric test. **E** smFISH validation of CAP-seq uniquely captured SG-proximal mRNA (*SMC1A*) and SG-excluded mRNA (*SRRM2*). Scale bars, 5 μm. **F** Quantitation of the ratio of smFISH within SG vs. whole cell. Bars and lines represent mean values and SDs, respectively (*n* = 20 cells from three independent experiments).

overlaps with SGs (Supplementary Fig. 9). Following RNA labeling, biotinylation signal is highly co-localized with both miniSOG and the SG marker, G3BP2 (Fig. 2A). We further confirmed that neither probe incubation nor blue light illumination affected SG formation (Fig. 2A). Differential expression analysis identified 2229 transcripts significantly enriched in post- versus pre-enrichment samples, while 1042 transcripts were significantly depleted (Supplementary Figure 10, Supplementary Data 2A). Comparison of transcript abundance enriched from G3BP1-miniSOG cell line versus untargeted miniSOG negative control yielded 2097 and 1201 significantly enriched and depleted transcripts, respectively (Fig. 2B, C, Supplementary Data 2A). Consistent with our previous analysis in HEK293T cells, the U-2 OS SG transcriptome is defined as the overlap of three DESeq2 datasets, yielding a final list of

1135 SG-proximal and 320 SG-excluded transcripts (Fig. 2C, Supplementary Data 2A).

We compared our SG CAP-seq datasets with those obtained by SG purification method (referred to as SG$_{coreRNA}$ by ref. 19). (Fig. 2D). Notably, among the 320 depleted transcripts in SG CAP-seq dataset, 24 have been previously identified as enriched, including 5 mitochondrial gene encoded RNAs. To resolve this discrepancy in RNA localization, we performed smFISH imaging of several CAP-seq depleted transcripts, including *SRRM2*, *PLXNB2*, *PKD1* and *MT-ND4* (Fig. 2E and Supplementary Fig. 11). Quantitation of smFISH images revealed that less than 10% of their RNA signal overlapped with the SG marker G3BP1, which confirmed their exclusion from the SG (Fig. 2F and Supplementary Fig. 11). It is likely

that these transcripts were contaminants introduced during SG purification.

Among 1135 CAP-seq enriched RNAs, 296 (26%) have also been identified as SG transcripts in the previous study, including *AHNAK*, *NORAD*, *PEG3*, *CDK6*, and *KDM5B*[12,19] were also found enrichment in the datasets. For the remaining 839 transcripts that are uniquely enriched in our SG CAP-seq in U-2 OS cells, 152 (18%) have also been identified in SG CAP-seq in HEK293T cells, which suggests that these are bona fide SG proximal transcripts. For example, *SMC1A* is undetectable in SG purification dataset but is significantly enriched by SG CAP-seq. Indeed, smFISH imaging analysis shows that more than half of the *SMC1A* mRNA signal (55 ± 6%) is localized within SG when cells are stressed with sodium arsenite treatment (Fig. 2E, F). Taken together, the above data demonstrate the higher spatial specificity and good coverage of CAP-seq as compared with traditional biochemical purification methods.

Previous studies have revealed a negative correlation between mRNA association with SG and translational efficiency (TE)[10,19,21,36]. Consistent with this view, our SG-proximal transcripts overall have modestly but significantly lower TEs than those of SG-excluded transcripts ($p = 0.04$, Mann-Whitney test) (Supplementary Fig. 12, Supplementary Data 2B). We also observed longer transcript length and higher AU content in SG-proximal versus SG-excluded mRNAs (Supplementary Fig. 12, Supplementary Data 2B), which follows a similar trend as previous reports that $SG_{coreRNA}$ contained less GC content[19]. The above analysis shows that data acquired with CAP-seq are generally in agreement with prior knowledge of SG transcriptome.

Notably, the molecular composition of SG is known to be cell-type specific. For example, TDP-43, the well-known SG protein marker in neuron, is not found in the SG proteome in Hela cells[23]. In addition, proteomic profiling experiments identified 123 and 411 SG proteins in arsenite-stressed HEK293T and U-2 OS cells, by APEX proximity labeling or purification of SG, respectively[18,20]. Among these, only 82 proteins were enriched in both datasets. It is, therefore expected that the SG transcriptome might also be cell-type specific. We compared the 457 and 1135 SG-proximal transcripts from arsenite-stressed HEK293T and U-2 OS cells, respectively. Among these, 261 transcripts are shared by both datasets (i.e., cell type-independent). The remaining 196 transcripts in the HEK293T SG CAP-seq dataset and 874 transcripts in the U-2 OS SG CAP-seq dataset are defined as HEK293T-specific and U-2 OS-specific, respectively (Supplementary Figure 13).

We compared the RNA features of cell type-specific and cell type-independent categories, but did not observe substantial differences in the translation efficiency, AU content, and transcript length. U-2 OS-specific CAP-seq RNAs tend to have higher AU content and shorter CDS length (Supplementary Fig. 14, Supplementary Data 2C). The trend is quite modest yet statistically significant. The above comparisons indicate that while the SG RNA compositions may vary substantially across different cell lines, presumably due to differences in the SG protein components that are responsible for recruiting these RNAs, the overall SG RNA features are quite similar, suggesting a conserved mechanism of RNA sorting into the SG.

### Analyzing stress-specific SG transcriptome

In addition to oxidative stress induced by sodium arsenite, the formation of SG could also be elicited by other means of stimulation that impairs translation initiation[1,32]. For instance, hyperosmotic shock with sorbitol has downregulated protein translation, ultimately leading to apoptosis[37]. As an intermediate in the polyol pathway, elevated sorbitol concentration also causes oxidative stress by decreasing the ratio of NAD⁺/NADH and inducing cellular ROS generation[38]. While both arsenite and sorbitol are capable of triggering the formation of SGs, the size and shape of granules appear to be stress-specific[39]. In terms of molecular components, smFISH imaging has revealed different trends of mRNA localization to SG when cells are challenged with either

sorbitol or ER stress[19]. Using purified RNA granules from NIH3T3 cells treated with ER stress, heat shock or sodium arsenite, previous research has identified both common RNA components and stress-dependent RNA targeting[10]. However, the RNA components of SG from different stimulation in human cells have yet to be studied at the transcriptome-wide level.

We treated G3BP1-miniSOG HEK293T cells with 0.4 M sorbitol for 150 min to induce SG formation. In replicated experiments, cells were then labeled with 10 mM PA and 15 min blue light illumination (Supplementary Fig. 15). HEK293T cells expressing untargeted miniSOG were used as the negative control. Immunofluorescence imaging showed co-localization between miniSOG and biotinylation signal (Fig. 3A). Notably, SGs induced by sorbitol appear less spherical and smaller than those induced by sodium arsenite. Differentially expression analysis of post- versus pre-enrichment samples identified 2870 and 1316 RNAs as significantly enriched or depleted, respectively (Supplementary Fig. 16, Supplementary Data 1B), while the analysis between G3BP1-targeted versus untargeted miniSOG yielded 1136 significantly enriched and 526 depleted RNAs (Fig. 3B, Supplementary Data 1B). Following the procedures in defining the transcriptome of sodium arsenite-induced SG, we finally identified 822 RNAs as SG-proximal and 285 as SG-excluded (Fig. 3C, Supplementary Fig. 17, Supplementary Data 1B) under sorbitol treatment.

We compared the SG-proximal transcriptomes induced by different stress. Among a total of 1136 SG RNAs, 143 (12.6%) are present in both stress conditions (stress-independent), whereas 679 (58.8%) and 314 (27.6%) are localized only in sorbitol-specific or arsenite-specific SGs, respectively (Fig. 3D). To verify our dataset, we performed smFISH imaging on several sorbitol-specific mRNAs (Fig. 3E). Co-localization analysis revealed almost 2-fold higher levels of SG targeting under sorbitol stress than arsenite stress for *MAD2L1* (75 ± 6% vs. 37 ± 9%), *PRDX3* (68 ± 7% vs. 39 ± 6%), and *MORF4L2* (73 ± 7% vs. 32 ± 8%) (Fig. 3F). To account for the differences in SG size and shape, we further quantified the ratios of FISH signal densities within versus outside the SGs. The ratios measured under sorbitol stress are significantly higher than those under the arsenite stress (*MAD2L1*, 29 ± 10 vs. 15 ± 6; *MORF4L2*, 43 ± 15 vs. 12 ± 6; *PRDX3*, 23 ± 6.2 vs. 14 ± 3.6) (Supplementary Fig. 18). Together, the above data demonstrate that CAP-seq could capture stress-specific SG transcripts with high spatial specificity.

To understand the mechanism underlying stress-specific RNA targeting to the SG, we analyzed the RNA features of our SG-proximal transcriptomes, including TE, transcript length, and m⁶A abundance (Supplementary Data 3C). No significant differences in TE and transcript length were found among mRNAs in stress-independent, sorbitol-specific and arsenite-specific datasets (Supplementary Fig. 19, Supplementary Data 3A–C). However, sorbitol-specific mRNAs overall are characterized with significantly shorter CDS and longer 3′ UTR (Fig. 3G, Supplementary Data 3C). In terms of m⁶A modification, arsenite-specific SG RNAs overall contain 28% higher numbers of m⁶A sites (weighed average) than sorbitol-specific SG RNAs (2.44 vs. 1.90 per RNA), while both are substantially higher than SG-excluded RNAs (Fig. 3H, Supplementary Data 3C). Recently, Jaffrey et al. reported that the m⁶A-binding protein YTHDF2 is more highly enriched in SGs induced by arsenite than in those induced by sorbitol[40]. In light of these observations, the enrichment of m⁶A sites and m⁶A-binding proteins appear highly correlated to each other, although the exact causal relationship has yet to be established.

### Mapping preexisting G3BP1–RNA interaction network

The dynamics of SG formation has been studied through both fluorescence microscopy[2] and proteomic analysis[22]. Super-resolution imaging has revealed the presence of miniature clusters of G3BP1 in unstressed cells[9]. Proximity labeling with BioID[23] or APEX[20] using SG core component as the bait has identified protein interaction networks that remain largely unchanged through stress stimulation, suggesting

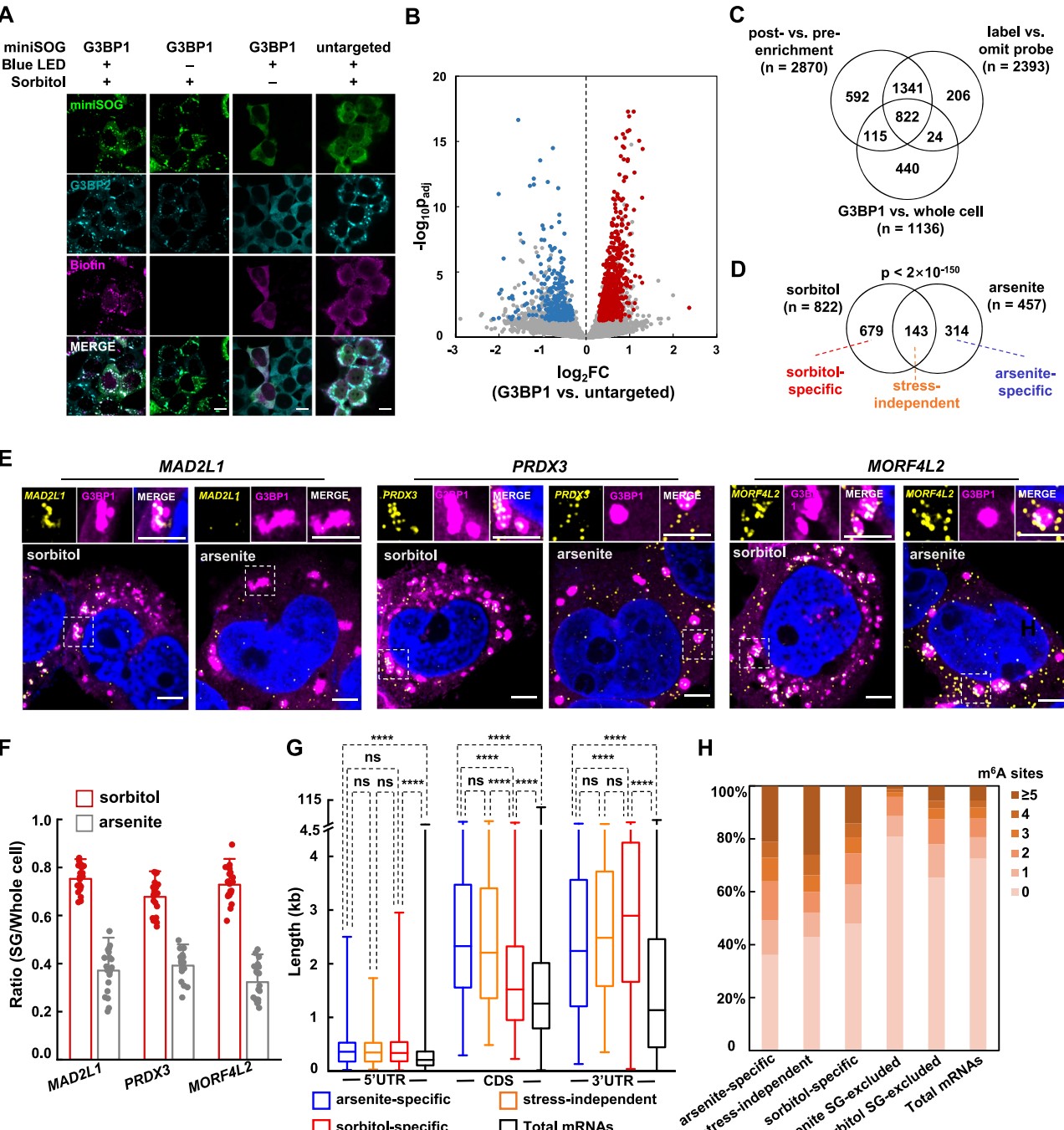

**Fig. 3 | Identifying stress-specific SG transcriptome. A** Immunofluorescence microscopy of HEK293T cells expressing G3BP1-miniSOG and untargeted miniSOG (green). Cells were treated with 0.4 M sorbitol for 150 min and labeled by 10 mM PA under 15 min blue light illumination. Endogenous SG marker G3BP2 and biotinylated signal are shown in cyan and magenta, respectively. Unstressed controls were performed at a separate time. Images are representative examples from three independent experiments. Scale bars, 10 μm. **B** Volcano plot comparing the abundance of RNA captured with G3BP1-miniSOG vs. untargeted miniSOG in CAP-seq labeled HEK293T cells under sorbitol treatment. Red and blue dots represent SG-proximal and SG-excluded transcripts, respectively. **C** Venn diagram comparing RNAs enriched in: (1) post- vs. pre-enrichment of G3BP1-miniSOG CAP-seq; (2) G3BP1-miniSOG labeling vs. negative control omitting PA; (3) G3BP1-miniSOG vs. untargeted miniSOG, SG-proximal RNAs in sorbitol treated HEK293T are defined as the overlap of the above three datasets. **D** Venn diagram comparing CAP-seq SG-proximal RNAs under arsenite and sorbitol stress. The $p$ value was calculated by hypergeometric test. **E** smFISH validation of stress-specific SG-proximal mRNAs. Scale bars, 5 μm. **F** Quantitation of the ratio of smFISH within SG vs. whole cell. Bars and lines represent mean values and SDs, respectively ($n = 20$ cells from three independent experiments). **G** Box plot comparing the length features between sorbitol-specific, stress-independent and arsenite-specific SG-proximal mRNAs. The box marks the first and third quartiles; whiskers indicate the minima and maxima; the central line represents the median. An unpaired Mann-Whitney U test (two-sided) was performed to evaluate the statistical significance. ns, not significant ($p > 0.05$); ****$p < 0.0001$. Transcript length features are referenced from Ensembl website. **H** Comparing the proportion of mRNA with different m6A sites between sorbitol-specific, stress-independent, arsenite-specific SG-proximal and SG-excluded datasets and Total mRNAs. m6A sites are referenced from a previous study[42].

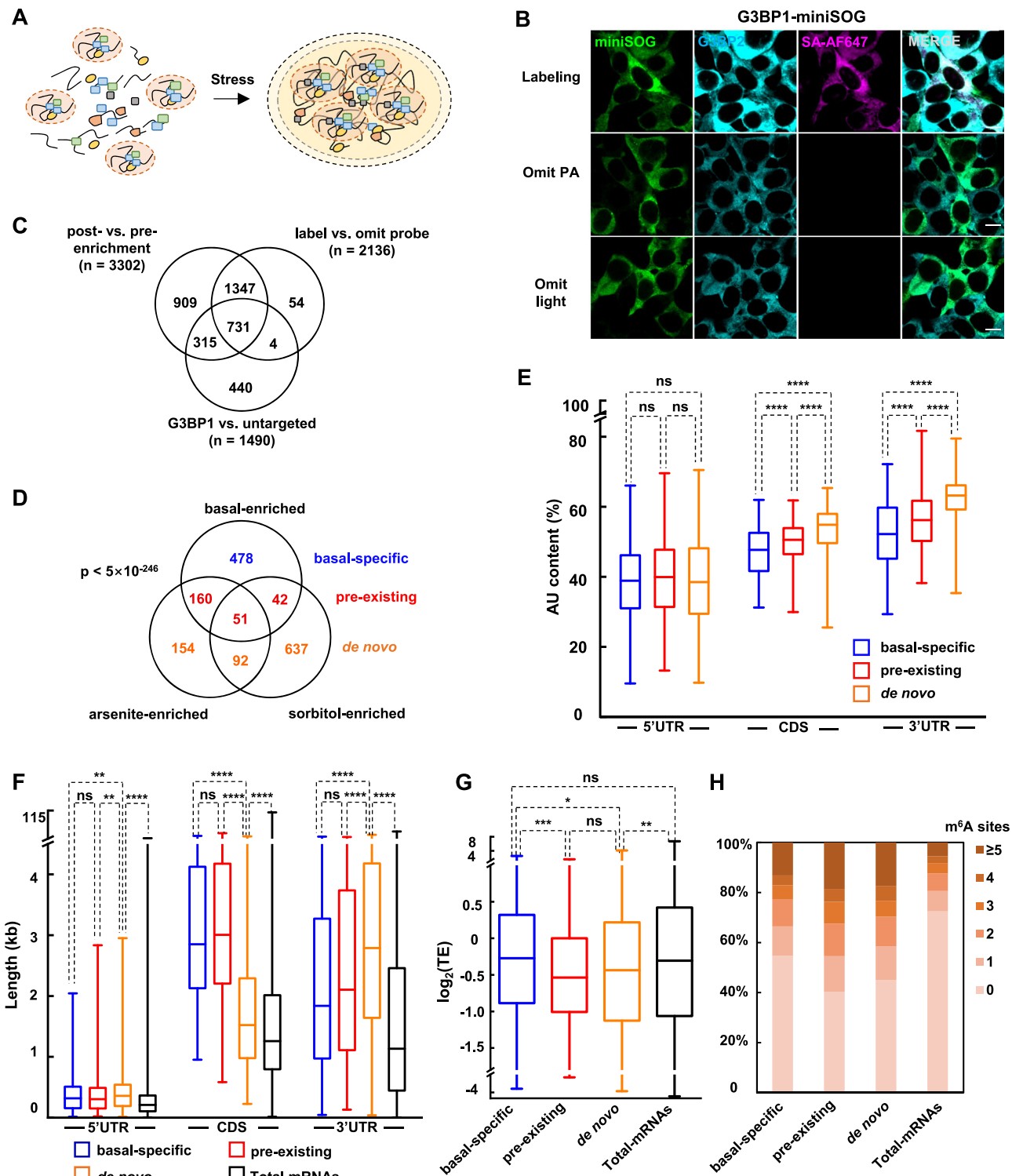

**Fig. 4 | Profiling pre-existing RNA interaction network in SG. A** Schematic of SG formation under stress. **B** Immunofluorescence microscopy of HEK293T cells expressing G3BP1-miniSOG and untargeted miniSOG (green). Cells were labeled by 10 mM PA under 15 min blue light illumination. Endogenous SG marker G3BP2 and biotinylated signal are shown in cyan and magenta, respectively. Images are representative examples from three independent experiments. Scale bars, 10 μm. **C** Venn diagram comparing RNAs enriched in: 1) post- versus pre-enrichment of RNA labeled with G3BP1-miniSOG; 2) RNA labeled with G3BP1-miniSOG versus RNA from negative control omitting PA; 3) RNA labeled with G3BP1-miniSOG versus untargeted miniSOG. G3BP1-proximal RNAs under unstressed conditions are defined as the overlap of the above three datasets. **D** Venn diagram comparing the CAP-seq enriched RNAs under unstressed, arsenite stress and sorbitol stress. The *p*

value was calculated by a hypergeometric test. Bar plot showing the comparison of AU content in different regions (**E**), length features (**F**) and translation efficiencies (**G**) between mRNAs in basal-specific, pre-existing and de novo datasets. The box marks the first and third quartiles; whiskers indicate the minima and maxima; the central line represents the median. An unpaired Mann-Whitney U test (two-sided) was performed to evaluate the statistical significance. ns, not significant ($p > 0.05$); *$p < 0.05$; **$p < 0.01$; ***$p < 0.001$; ****$p < 0.0001$. Translation efficiencies are counted from a previous report[56]. Transcript length features and AU content are referenced from Ensembl website. **H** Comparing the proportion of mRNA with different m6A sites in basal-specific, preexisting and de novo datasets. m6A sites are referenced from a previous study[41].

the presence of pre-existing interaction within the SG structure. The above studies have favored a view in which nanoscopic 'seeds' of pre-existing biomolecular interactions grow and fuse upon stress stimulation, while recruiting more components to mature into microscopically visible granules (Fig. 4A). Probing the pre-existing interaction network could thus provide key information on the formation and regulation of SG during stress response. However, fractionation-based methods are not suited for such analysis. While the protein interaction networks have been investigated with proximity labeling[20,23], pre-existing RNA interaction networks has remained unsolved.

To probe pre-existing interaction networks surrounding SG core components, we performed CAP-seq in G3BP1-miniSOG HEK293T cells in the absence of stress. HEK293T cells expressing untargeted miniSOG were used as the negative control. Immunofluorescence imaging showed the co-localization between biotinylated signal and miniSOG (Fig. 4B). Following the same DESeq2 work flow and cutoff values ($\log_2 FC > 0.3$ and $p_{adj} < 0.05$) as in the previous analysis of stressed cells, CAP-seq identifies 3302, 2136, and 1490 transcripts in post- vs. pre-enrichment, labeling vs. omitting PA, and G3BP1-miniSOG vs. untargeted miniSOG analyses, respectively. The overlap of these three datasets yields 731 RNAs, which we define as G3BP1-proximal transcripts under the basal condition (Fig. 4C, Supplementary Fig. 20, Supplementary Data 1C). Our analysis also reveals 218 transcripts that are significantly depleted between post- vs. pre-enrichment and between G3BP1-targeted vs. untargeted miniSOG samples, which we define as G3BP1-excluded transcripts at the basal level (Supplementary Fig. 21, Supplementary Data 1C).

We compared the list of G3BP1-proximal transcripts under basal condition against SG-proximal transcriptomes under arsenite and sorbitol stress. Among a total of 1614 transcripts, 478 RNAs are exclusively enriched in unstressed G3BP1-proximal dataset, which we define as basal-specific transcriptome (Fig. 4D). These RNAs appear proximal to G3BP1 only at the basal level and are no longer SG-localized upon stress stimulation. A total of 253 RNAs are captured in both unstressed G3BP1-proximal and stressed SG-proximal transcriptome with at least one type of stress, which we define as pre-existing SG transcriptome. Among these, 51 RNAs (20%) are enriched in both arsenite-induced and sorbitol-induced SG-proximal transcriptome (Fig. 4D). The remaining 883 RNAs, defined as de novo SG transcriptome, are captured with G3BP1 CAP-seq only under stressed conditions, among which 92 (10%) are present in both types of stress stimulation.

RNA feature analysis reveals significantly higher AU content in both pre-existing and de novo SG transcriptomes than in basal-specific or the total cellular transcriptomes, particularly for the CDS and 3′ UTR (Fig. 4E, Supplementary Data 3C). In contrast, no significant difference in AU content was observed in the 5′ UTR. In terms of transcript length, RNAs in all three G3BP1-/SG-proximal datasets are overall longer than total cellular mRNAs (Supplementary Fig. 22, Supplementary Data 3B-C), which is in accordance with the previous observation that SG-localized mRNAs are longer than average[10,19]. While the basal-specific and pre-existing SG transcriptomes are characterized with longer CDS length, RNAs from de novo SG transcriptome contain significantly longer 3′ UTR and 5′ UTR (Fig. 4F, Supplementary Data 3C). We also found significantly lower TE for RNAs in pre-existing and de novo SG transcriptome, as compared to the basal-specific dataset or the total mRNAs (Fig. 4G, Supplementary Data 3C).

RNA methylation, such as m[6]A, has been found to enhance the ability of RNA to participate in liquid-liquid phase separation[8]. Indeed, m[6]A reader proteins YTHDF1/2/3 have been shown to localize at SGs upon sodium arsenite stress[8,9,20]. We thus analyzed the number of m[6]A sites and m[6]A site density (number per kilobase) in our datasets, using published m[6]A database in HEK293T cells[41]. Both pre-existing and de novo SG-transcriptomes contain higher percentage of m[6]A-modified

RNAs than the basal-specific transcriptome, indicating a positive correlation between SG assembly and m[6]A RNA recruitment (Fig. 4H, Supplementary Figure 23, Supplementary Data 3C). This observation is consistent with the previous report that RNAs in SGs contain more m[6]A modifications than the total RNA pool[8,9].

## Mapping SG transcriptome during granule disassembly

As exemplified by SGs, membraneless organelles often feature a highly dynamic and sometimes reversible assembly process in the cellular environment. Just as SG formation can be rapidly triggered by external stimuli within 1 h, upon removal of the stress, SGs disintegrate on the time scale of hours. It has been postulated that SG disassembly may be an ordered process[2], with sub-populations of RNAs dissolving into the cytoplasm faster and thus functioning earlier (e.g., re-entry into protein translation) than the rest of the SG transcriptome. This raises the interesting questions of what and to what extent do RNA features influence the disassembly dynamics. Answers to these questions would greatly advance our understanding of how cells orchestrate biomolecular assembly in response to metabolic challenges such as oxidative stress. Yet, it has remained technically difficult to perform such analysis.

We leveraged the precise temporal control and 10-min-level temporal resolution of photo-activatable CAP-seq to profile SG-proximal transcriptome during the granule disassembly process. Following 1 h of oxidative stress, sodium arsenite was removed from the cell culture media. As sodium arsenite-induced eIF2α phosphorylation mediates the inhibition of translation initiation[32], we monitored eIF2α phosphorylation levels with western blot during the recovery phase, which gradually decreased upon removal of stress during the time course of 3 h (Supplementary Fig. 2). Notably, at 1 h post-stress, the phosphorylation level of eIF2α remains higher than the basal level, indicating persistent translational repression at this time point. Consistent with the western blot analysis, confocal fluorescence imaging of HEK293T cells also reveals a gradual disassembly of SGs during the recovery, with the sizes of granules slightly reduced at 1 h post-stress (Fig. 5A). At 3 h post-stress, the eIF2α phosphorylation appears fully restored to its basal level (Supplementary Fig. 2), and no microscopically visible SGs can be observed in most cells (Fig. 5A). Indeed, smFISH imaging of two highly enriched SG mRNAs in our datasets, APLP2 and GAS1, are gradually dispersed throughout the cytoplasm as SGs disassemble during the recovery phase (Fig. 5B).

We thus chose 1 h (T1) and 3 h (T3) post-stress as the time points for CAP-seq labeling of SGs during the recovery phase. HEK293T cells expressing untargeted miniSOG are used as the negative control. Following the identical RNA labeling protocol (10 mM PA and 15 min blue LED illumination), DESeq2 work flow (post- vs. pre-enrichment, labeling vs. omitting PA, and G3BP1-miniSOG vs. untargeted miniSOG) and cutoff values ($\log_2 FC > 0.3$ and $p_{adj} < 0.05$) as in the previous analyses of stressed and unstressed cells, a total of 533 and 428 G3BP1-proximal transcripts were captured at T1 and T3, respectively (Supplementary Fig. 24, Supplementary Data 1D, E). We performed hierarchical clustering analysis to the union of G3BP1 CAP-seq datasets across T0, T1 and T3 time points (Supplementary Fig. 25). A total of 949 transcripts are clustered into four categories based on their $\log_2 FC$ (G3BP1 vs. untargeted) values (Fig. 5C). Each cluster represents a distinct pattern of relative enrichment during the disassembly phase. For example, mRNAs in cluster 1 are highly enriched in SG at the onset of recovery, but rapidly dissociate from the granule within ~1 h. In contrast, mRNAs in cluster 2 dissociate more slowly during the recovery phase. Clusters 3 and 4 represent mRNAs that are gradually more enriched to the vicinity of G3BP1 despite granule disassembly, even when SG foci are no longer visible at T3.

We sought to understand the sorting mechanism underlying differential retention/dispersion of mRNA with respect to the granule. Previous research has suggested that RNA could resume translation

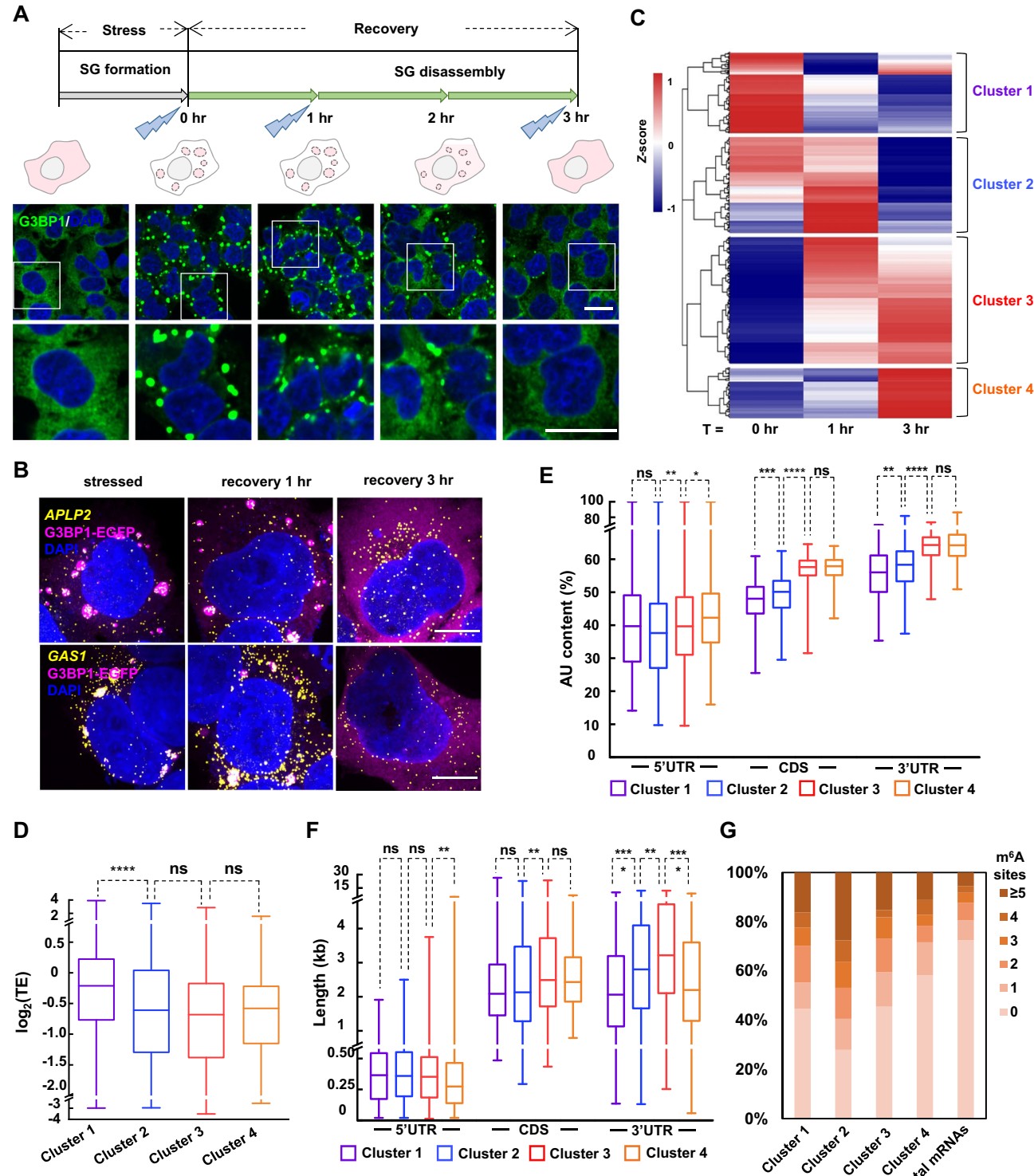

**Fig. 5 | Resolving the kinetics of RNA dissociation during SG disassembly.**
**A** Schematic and immunofluorescence imaging of the dynamic changing of SG disassembly after washing out of arsenite stress in G3BP1-miniSOG HEK293T. Scale bars, 20 μm. **B** smFISH visualizing the dispersion of SG-proximal mRNAs (*APLP2*, *GAS1*) after washing out of arsenite stress. Scale bars, 10 μm. **C** Heat map showing mRNAs presented different patterns of enrichment during disassembly based on hierarchical clustering analysis to the union of G3BP1 CAP-seq datasets under T0, T1 and T3. Bar plot showing the comparison of mRNAs translation efficiencies (**D**), AU content of different regions (**E**) and length features (**F**) between 4 clusters. The box

marks the first and third quartiles; whiskers indicate the minima and maxima; the central line represents the median. An unpaired Mann-Whitney test (two-tailed) was performed to evaluate the statistical significance. ns, not significant ($p > 0.05$); * $p < 0.05$; ** $p < 0.01$; *** $p < 0.001$; **** $p < 0.0001$. Translation efficiencies are counted from a previous report[56]. Transcript length features and AU content are referenced from Ensembl website. **G** Comparing the proportion of mRNA with different m6A sites between four clusters. m6A sites are referenced from a previous study[41]. Images are representative examples from three independent experiments.

following SG disassembly[12]. In line with the above observations, our RNA feature analysis has revealed a correlation between TE and RNA dissociation from the SG. The median TE of mRNAs in cluster 1 is approximately 32% higher than those of mRNAs belonging to other clusters (Fig. 5D, Supplementary Data 3D). mRNAs in cluster 1 and 2 are characterized with lower AU content, especially in their CDS and 3′ UTR regions. (Fig. 5E, Supplementary Data 3D). In terms of transcript length, RNAs in clusters 2 and 3 share the feature of longer 3′ UTR (Fig. 5F, Supplementary Data 3D), indicating that longer transcripts have a stronger tendency to interact with SG components during the early stage of recovery. Interestingly, cluster 4 is characterized with both high AU content and short RNA length in the CDS and 3′ UTR, which suggests that AU-rich elements might play a major role in RNA resident in SG at the final stage of stress recovery.

It has been shown for SG localized mRNAs that a higher level of $m^6A$ modification is correlated with more rapid re-entry into translation[42]. We, therefore compared the extent of $m^6A$ sites and $m^6A$ site density (number per kilobase) in our dataset. Overall, mRNAs across the four clusters contain substantially more $m^6A$ sites than the total cellular mRNA population. Among these, cluster 2 has higher ratio of $m^6A$ mRNA than cluster 1, indicating a negative correlation between $m^6A$ sites with mRNA dissociation rate from the SG (Fig. 5G, Supplementary Fig. 26, Supplementary Data 3D).

While both confocal fluorescence imaging of SG marker proteins and western blot analysis of eIF2α phosphorylation indicated successful cellular recovery from stress following the removal of arsenite for 3 h (Fig. 5A, Supplementary Fig. 2), we were curious whether the G3BP1-proximal transcriptome has been reset to the initial unstressed state at this time point. Comparison of our G3BP1 CAP-seq datasets collected at T3 versus at basal condition reveals an overlap of 126 RNAs, while 302 and 605 RNAs were uniquely found at T3 and under basal level, respectively (Fig. 6A). It appears that a sub-population of mRNAs remain proximal to G3BP1 in the cytoplasm despite apparent SG disassembly. Notably, compared to G3BP1-proximal transcripts at the basal level, RNAs captured at T3 are characterized with higher AU content in the CDS and 3′ UTR (Fig. 6B), shorter CDS length, longer 3′ UTR length (Fig. 6C), and lower TEs (Fig. 6D), a trend that is reminiscent to the comparisons between G3BP1-proximal transcripts captured at the stressed versus the basal levels (Fig. 4E−G). Taken together, the above analysis favors the view that cells contain G3BP1-centered nanostructures, which are strengthened during stress challenge, and which are memorized long after the removal of stress (Fig. 6E). Such structures may serve as the hubs of RNA sequestration and/or translational control, which could be leveraged by cells to better cope with external challenges.

## Discussion

To summarize, we have applied CAP-seq to profile SG-proximal transcriptome during various stages of assembly/disassembly in cultured mammalian cells. Compared to fractionation-based methods, CAP-seq offers higher spatial specificity as demonstrated with smFISH imaging. For example, mRNAs *SRRM2*, *PLXNB2*, *PKD1*, and *MT-ND4*, were previously enriched from purified SG[19] but were found to be SG-excluded by CAP-seq. Our quantitative smFISH imaging analysis shows less than 10% co-localization of these RNAs with SG, thus supporting the CAP-seq dataset. We also compared our SG-proximal datasets with RNAs enriched in granules isolated from arsenite-stressed NIH-3T3 cells[10], revealing an overlap of 181 transcripts. Notably, both studies identified a positive correlation between transcript length and SG enrichment. Such correlation has also been reported by previous research on liquid-liquid phase separation of yeast RNAs in vitro[7] and RNA granules purified from arsenite-stressed U-2 OS cells[36].

We also compared our CAP-seq dataset with a previously reported SG transcriptome profiling with APEX-seq, in which the translation factor eIF4A was used as the bait and SG formation was induced by

heat-shock for 20 min in HEK293T cells[21]. By applying the same cutoff as in our analysis ($\log_2FC > 0.3$ and $p_{adj} < 0.05$), differential expression analysis of the APEX-seq dataset reveals 394 transcripts as significant enriched. Among these, only 18 transcripts overlap with our CAP-seq dataset of 457 transcripts, which may be attributed to the differences in the choices of baits, stress conditions, and durations of stress. Nevertheless, both datasets identified a strong correlation between SG-proximal RNA labeling and transcript length, suggesting that longer RNAs are more likely localized to the SG.

More recently, Kleiner and co-workers applied proximity-dependent RNA editing technique (TRIBE-ID) to profile G3BP1-associated RNAs in HEK293T cells challenged with sodium arsenite stress[43]. Among the 745 G3BP1-associated RNAs identified in their dataset, 106 were enriched in our CAP-seq SG RNA datasets derived from arsenite stressed HEK293T (p value $< 4 \times 10^{-100}$). Notably, their data revealed that G3BP1-associated RNAs are positively correlated with their transcript length and negatively correlated with translation efficiency, which is consistent with our findings.

Our dataset revealed that while both arsenite and sorbitol stress could lead to the assembly of SGs, their transcriptome contents differ substantially. It has been shown that cells with double knockout of G3BP1/2 lost the ability to form SG under arsenite stress, but could still form SG-like foci when treated with sorbitol[39]. In addition, sorbitol and arsenite have exhibited opposite effects on the activity of p70 S6 kinase and 4E-BP1, both of which are major regulators of protein synthesis in the mTOR pathway[44]. Thus, the observed stress-specific SG-proximal transcriptome may arise from the distinct molecular pathways triggered by a specific type of stress.

The molecular organization of SGs has been described in a two-layer model: a dense, stable and purifiable "core" enclosed by a dynamic "shell" that frequently exchanges components with the cellular environment[1,18]. Live cell fluorescence imaging of SG assembly has demonstrated that stress triggers the nucleation of G3BP1-associated cores, which gradually undergo fusion and growth into mature SGs[2]. Proximity-labeling experiments have also portrayed an extensive protein-protein interaction network formed by SG core components even in the absence of stress[20,23]. Since G3BP1 is an RNA binding protein, it is expected to interact with mRNAs in both stressed and unstressed states. In this study, we leveraged the high spatial specificity of CAP-seq to resolve G3BP1-proximal RNA interaction network at unstressed condition, which is not attainable with previous fractionation-based approaches. Our data reveal that mRNAs with longer and more AU-rich 3′ UTR are preferentially recruited to SGs.

One of the most prominent features of membraneless organelles is the reversibility of their formation. Live cell imaging of the SG disassembly has revealed the non-uniform SG disintegration into smaller foci before undergoing further clearance[2]. In this study, we applied CAP-seq to investigate G3BP1-proximal transcripts during SG disassembly. Our data indicate that longer CDS and 3′ UTR might contribute to a stronger potential to participate in RNA-protein and RNA-RNA interactions for maintaining the residual nanoscopic SG structure. We also observed higher TE in G3BP1-proximal transcripts captured during arsenite stress than in those captured during recovery. Our observation is consistent with the recent study of native peptides captured during stress recovery stages, which showed that SG-localized mRNAs tend to re-enter translation more rapidly[45]. In addition, our data revealed that AU content, especially in 3′ UTR sequence, might play a central role in sorting mRNAs to associate with G3BP1 during the final stage of disassembly.

While SGs are typically invisible in most cells after 3 h of recovery, it remains unknown whether the RNA interaction network has been reset to the basal state. Our observation that a sub-population of mRNAs appears to remain proximal to G3BP1 may likely arise from interactions with SG core components, although we do not have experimental data to directly support this speculation. Future super-resolution imaging

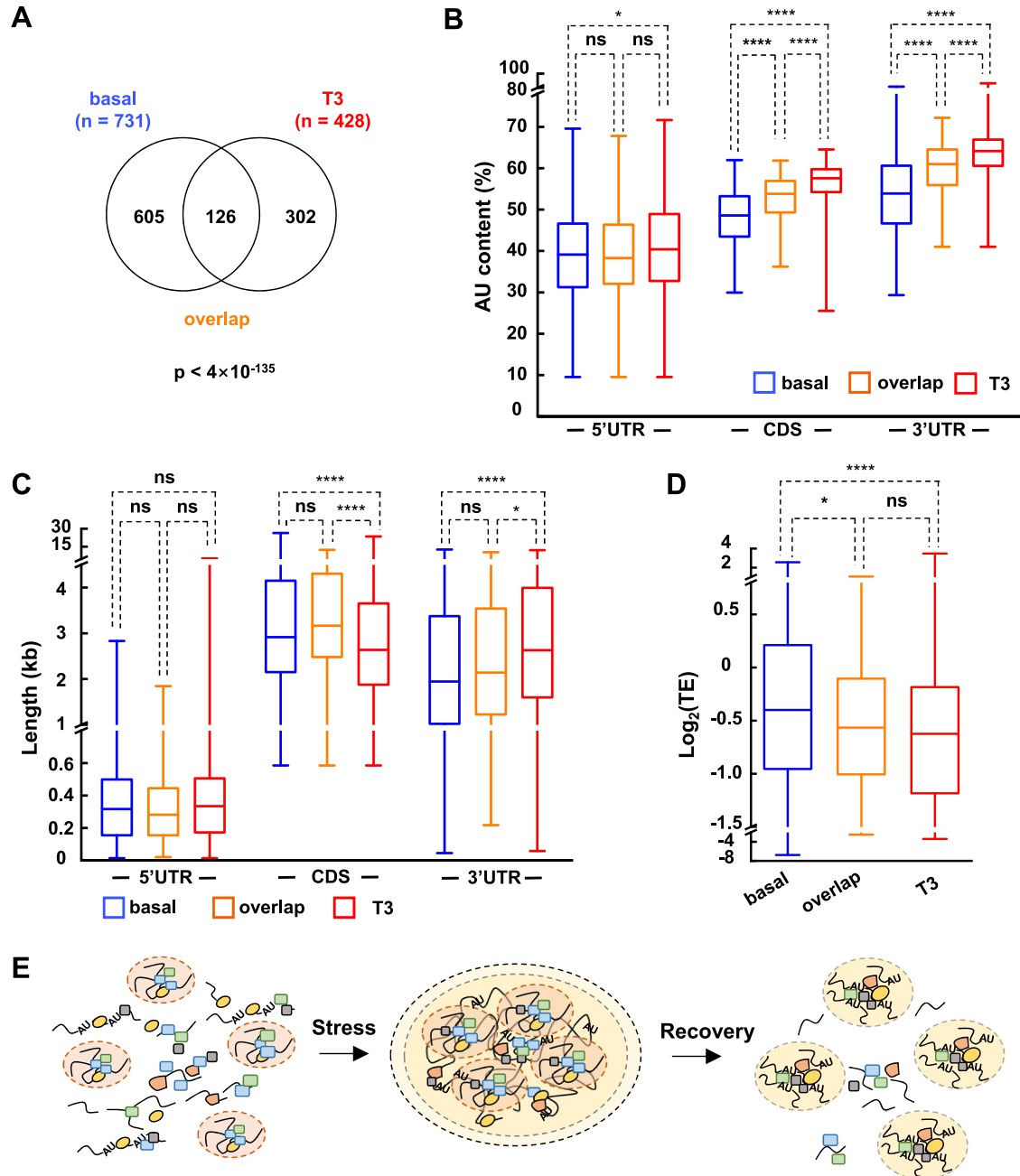

**Fig. 6 | Comparing G3BP1-proximal RNAs at basal level versus post-stress recovery. A** Venn diagram comparing transcripts enriched at basal level versus at T3. The p values were calculated by hypergeometric test. Bar plots comparing the AU content (**B**), length features (**C**) and translation efficiencies (**D**) between mRNAs in basal and T3 datasets. The box marks the first and third quartiles; whiskers indicate the minima and maxima; the central line represents the median. An unpaired Mann-Whitney U test (two-sided) was performed to evaluate the statistical significance. ns, not significant ($p > 0.05$); * $p < 0.05$; ** $p < 0.01$; *** $p < 0.001$; **** $p < 0.0001$. Translation efficiencies are provided by a previous report[56]. Transcript length features and AU content are referenced from Ensembl website. **E** Schematic model of stress-induced reconstruction of G3BP1-centered nanostructure following SG disassembly.

experiments may provide further evidence for resolving this issue. Furthermore, our analysis reveals that comparisons of RNA features between transcripts enriched under basal and T3 are reminiscent to the comparisons between G3BP1-proximal transcripts captured at the stressed versus the basal levels, suggesting that the G3BP1-related core are memorized long after the removal of stress and served as the hubs of RNA sequestration and/or translational control.

It should be noted that RNA features such as AU content have been reported to be correlated with transcript length[46]. To better examine the correlation between SG enrichment with total RNA length and AU content at the transcript level, we broke down the transcripts into 5' UTR, CDS, and 3' UTR segments. We observed quite different trends at each segment. For example, stress-de novo RNAs feature both longer length and higher AU content in their 3' UTR, whereas their CDSs are shorter with higher AU content (Fig. 4E, F). RNAs remained proximal to G3BP1 during final stages of SG disassembly (cluster 4) were also characterized with shorter 3' UTR but more AU. These observations indicate that AU content and length may contribute differently to SG enrichment.

Several lines of evidence have suggested that m[6]A modification plays a key role in RNA localization to SGs[8,9]. Recently, Jaffrey and co-workers showed that m[6]A-binding protein YTHDF2 was more highly

enriched in arsenite-induced than sorbitol-induced SGs[40]. Consistent with these findings, we observed more m6A sites in SG-proximal RNAs under arsenite stress. Future study that combines proteomic and transcriptomic data in the same cell line with the same bait protein could help elaborate the relationship between m6A-mRNA enrichment and m6A-binding protein localization during various stages of SG assembly/disassembly. Such multipronged approach would require photoactivatable protein labeling with miniSOG, which has been recently demonstrated by us and other groups[47–49].

Finally, this study is limited by mild cytotoxicity, insufficient temporal resolution, and potential bias towards exposed RNAs. First, prolonged treatment of PA causes cellular stress, as revealed by elevated eIF2α phosphorylation. Although the level of stress has not reached the threshold of inducing SG assembly and could be normalized by using appropriate controls, it might still impede the application of CAP-seq to more sensitive and delicate cells such as primary neuron cultures. Second, CAP-seq labeling is potentially biased towards prey RNAs that are more solvent exposed, such that the reactive singlet oxygen could access guanosine nucleobase for successful labeling. Notably, this bias is shared by other proximity labeling methods, such as APEX2[25], BioID[50]/TurboID[51], and RinID[48]. Finally, CAP-seq requires blue LED illumination for 15 min to achieve efficient RNA labeling. This temporal resolution is still insufficient for resolving SG components during its rapid assembly stage, which typically matures within 15–30 min. In the future, more effective tools for photosensitive catalysis could be developed by directed evolution.

## Methods

### Reagents
Information on the reagents, antibodies, and plasmids used in this work can be found in the Key Resource Table in Supplementary Information.

### Mammalian cell culture
HEK293T and HEK293T/17 cells were cultured in complete medium consisting of Dulbecco's Modified Eagle's Medium (DMEM, Gibco, C11995500BT) supplemented with 10% fetal bovine serum (Gibco, 10099141 C) at 37 °C with 5% CO2. U-2 OS cells were cultured in McCoy's 5A medium (BI, 01-075-1ACS) supplemented with 10% fetal bovine serum at 37 °C with 5% CO2.

To prepare lentivirus, HEK293T/17 cells were cultured in six-well plates to 60% confluency and transfected with pLX304 lentiviral vector encoding the genes of interest (1 μg) and two packaging plasmids dR8.91 (1 μg) and pVSV-G (700 ng) mixed with 5.4 μL Lipofectamine 2000 (Invitrogen, 10099141) in 200 μL Opti-MEM (Gibco, 31985062), following the manufacturer's instructions. After the medium was changed to DMEM, the mixture were carefully added to the cells. After 4–8 h incubation at 37 °C with 5% CO2, the medium was replaced by fresh complete medium. Lentivirus was collected 48 hours later, passed through a 0.45 μM disc filter to remove cell debris, flash frozen in liquid nitrogen, and then stored at −80 °C. Lentivirus packaged with miniSOG-NES sequence was prepared by Heyuan Biotechnology (Shanghai).

HEK293T cells and U-2 OS cells were cultured in 6-well plates to reach ~70% confluency (~600,000 HEK293T cells and 400,000 U-2 OS cells) when they were infected by lentiviruses. For lentivirus encoding G3BP1-miniSOG or G3BP1-EGFP, 1 mL lentivirus suspension was added to cells. For lentiviruses encoding untargeted miniSOG, HEK293T and U-2 OS cells were incubated with complete medium containing 5 μg/mL polybrene (provided by Heyuan Biotechnology) before infected with lentivirus at MOI 20 and MOI 40, respectively. After 48 hours of infection, cells were cultured in complete medium with 5 μg/mL blasticidin (Selleck, S7417) for 7 days to screen for cells stably expressing the fusion protein. Thereafter, cells expressing miniSOG were enriched with fluorescence activated cell sorting. miniSOG expression in each cell line was verified by immunofluorescence against the V5 tag.

### CAP-seq labeling in live cells
HEK293T and U-2 OS cells stably expressing G3BP1-miniSOG and untargeted miniSOG were cultured to 90% confluency in 15-cm culture dishes. To induce stress, cells were incubated in complete media supplemented with either 0.5 mM sodium arsenite (Sigma, S7400; stored as 500 mM aqueous stock) for 40 min or 0.4 M D-sorbitol for 130 min at 37 °C. Cells were then washed once with 1× HBSS and the medium was replaced by HBSS containing 10 mM PA (Accela, SY002930, stored as 1 M aqueous stock) and 0.5 mM sodium arsenite or 0.4 M sorbitol. For unstressed samples, cells were rinsed with HBSS once and then incubated in HBSS containing 10 mM PA. After 5 min of incubation under 37 °C, cells were illuminated with blue LED (emission peak 465-475 nm, 24 mW/cm²) for 15 min at room temperature.

For detection of SG transcriptome during disassembly, HEK293T cells stably expressing G3BP1-miniSOG or untargeted miniSOG were cultured to 90% confluency in 10-cm dishes and incubated with complete medium containing 0.5 mM sodium arsenite for 1 h. Thereafter, stressed cells were washed three times with pre-warmed complete medium at 37 °C. After recovering in complete medium for 40 min or 160 min, cells were washed once with HBSS, incubated with HBSS in the presence of 10 mM PA for 5 min at 37 °C, and illuminated with blue LED for 15 min at room temperature.

Following CAP-seq labeling, cells were washed twice with PBS before lysed with TRIzol Reagent (Invitrogen, 15596018). RNA was extract following the manufacturer's instructions (~700–1000 μg from HEK293T cells in a 15-cm dish), digested with 1-2 μL DNaseI (NEB, M0303) at 37 °C for 30 min, and then incubated with click reagents consisting of 0.1 mM biotin-azide, 2 mM THPTA, 0.5 mM CuSO4 and 5 mM sodium ascorbate. After 10 min of CuAAC reaction at room temperature, RNAs were purified with RNA Clean&Concentrator kit (Zymo Research, R1019) and eluted with pre-warmed nuclease-free water. The RNA integrity was detected by Fragment Analyzer (Agilent). Approximately 1 μg RNA was kept as pre-enrichment samples, while the remaining was purified with streptavidin beads.

### Immunofluorescence
HEK293T cells stably expressing G3BP1-miniSOG and untargeted miniSOG were cultured on glass coverslips pre-coated with Matrigel. The introduction of stress and CAP-seq labeling were performed as described above. Cells were then washed with PBS once, fixed by 4% (m/v) paraformaldehyde in PBS for 15 min at room temperature, and permeabilized by 0.1% Triton-100 (Sigma, T8787) in PBS. For click reaction, fixed and permeabilized cells were incubated with 50 μM N3-PEG3-biotin (100 mM stock in DMSO), 2 mM CuSO4, 1 mM BTTAA and 0.5 mg/mL sodium ascorbate in PBS at room temperature for 30 min. Thereafter, cells were washed with PBS for three times and blocked by 3% (w/v) BSA in PBS for 1 h at room temperature.

For immunostaining, cell samples were incubated with primary antibodies (mouse-anti-V5, Biodragon, 1:1000 dilution; anti-TIA1, abcam, 1:200 dilution; rabbit-anti-G3BP2, abcam, 1:200 dilution) at room temperature for 1 h or at 4 °C overnight. After washed with PBS for three times, cells were incubated with fluorophore-conjugated secondary antibodies (goat anti-mouse-Alexa Fluor 488, Thermo-Fisher, 1:1000 dilution; goat anti-rabbit-Alexa Fluor 647, Thermo-Fisher, 1:1000 dilution) and streptavidin-Alexa Fluor 568 (ThermoFisher, 1:2000 dilution) for 40 min. Labeled cells were then washed with PBS for three times, counterstained with DAPI (Thermo-Fisher, D1306) for 10 min at room temperature, and washed again three times with PBS before imaging.

### Western blot
HEK293T and U-2 OS cells stably expressing G3BP1-miniSOG were cultured in six-well plates to 80–90% confluency. Cells were washed in PBS and lysed by ultrasonication on ice in RIPA buffer (CWBio, CW2334S) containing 1 × protease inhibitor cocktail (Roche,

04693132001) and 1× Phosphatase Inhibitor Cocktail (bimake, B15001). Lysates were then diluted in 1× loading buffer (CWBio, CW0027S) and heated at 95 °C for 10 min, followed by centrifugation at 12,000 g for 10 min at 4 °C. Following electrophoresis in 10% Bis-Tris gels, protein bands were transferred to nitrocellulose membranes. The blots were blocked with 3% BSA in TBST (50 mM Tris, 150 mM NaCl, 0.1% Tween-20, pH 7.4-7.6) for 1 h at room temperature and then incubated with primary antibodies (rabbit-anti-G3BP, Abcam, 1:2000 dilution; rabbit-anti-eIF2α, CST, 1:1000 dilution; rabbit-anti-EIF2S1 (phosphor S51), abcam, 1:1,000 dilution) at 4 °C overnight. The blots were washed three times with TBST and incubated with secondary antibodies (Rabbit Anti-Goat IgG H&L (HRP), Biodragon, 1:4000 dilution). Chemiluminescence was performed by incubating the blot with Clarity Western ECL Substrate (Bio-Rad, 1705061) and imaged on a ChemiDoc imaging system (Bio-Rad).

## RNA affinity purification

20 - 40 μL of Dynabeads MyOne Streptavidin C1 (Life Technologies, 65002) were washed three times with 200 μL B&W buffer (5 mM Tris pH 7.5, 1 M NaCl, 0.5 mM EDTA, 0.1% (v/v) Tween-20), twice with solution A (0.1 M NaOH, 0.05 M NaCl in nuclease-free water), once with solution B (0.1 M NaCl in nuclease-free water), and re-suspended in 200 μL blocking buffer (1 mg/mL BSA, 1 mg/mL Yeast-tRNA in B&W buffer) on a shaker (1200 rpm) for 2 h at 25 °C. Thereafter, pre-blocked beads were washed three times with 200 μL B&W buffer. Extracted RNAs were mixed with an equal volume of 2 × B&W buffer (10 mM Tris pH 7.5, 2 M NaCl, 1 mM EDTA, 0.2% (v/v) Tween-20) before incubating with the preblocked beads on a shaker (1200 rpm) for 40 min at 25 °C to allow binding of biotinylated RNAs. The supernatant was discarded and the beads were washed three times with 200 μL B&W buffer, twice with 200 μL Urea buffer (4 M Urea, 0.1% (w/v) SDS in PBS), and twice with 200 μL PBS. The beads were finally re-suspended in 50 μL Elution buffer (95% formamide, 10 mM EDTA, 1.5 mM D-biotin), heated first at 50 °C for 5 min and then at 90 °C for 5 min. The supernatant containing eluted biotinylated RNAs was transferred to a 1.5 mL Eppendorf tube and mixed with 1 mL TRIzol reagent to extract RNA. Each sample was added with 200 μL chloroform and mixed vigorously, followed by centrifugation for 15 min at 4 °C, 12,000 g. The aqueous phase with dissolved RNAs was transferred to a new tube, added with 500 μL iso-propanol and 20 μg glycogen (RNA grade, Fermentas, R0551), and then incubated at -20 °C overnight to precipitate RNA. The sediment was washed with 1 mL 75% (v/v) ethanol and dissolved into 10 - 20 μL of nuclease-free water.

All RNA-related experiments were performed in a benchtop RNA work station and handled by RNase-free tubes and tips.

## Library construction

100 ng pre-enrichment and 5 μL post-enrichment (label or omitting probe) RNAs were used for cDNA library construction with NEBNext Ultra II RNA Library Prep Kit for Illumina (NEB, E7770). Total RNAs were fragmented to ~300 nt in the presence of 1 μL Random primer and 4 μL First Strand Synthesis Reaction Buffer at 94 °C for 9 - 12 min (depending on the RQN of pre-enrichment samples). Then the fragmented RNAs were reverse transcribed with the addition of 2 μL First Strand Synthesis Enzyme Mix and 8 μL nuclease-free water, followed by incubating at 25 °C for 10 min, 42 °C for 25 min, and 70 °C for 15 min. Second Strand Synthesis was preformed after the reaction. The mixtures were purified with 1.8× VAHTS DNA Clean Beads (Vazyme, N411) and eluted by 50 μL 0.1× TE buffer (Sigma, 93283) at room temperature. The double-stranded DNAs were then reacted with End Prep Eeaction and Adaptor Ligation, followed by purification with 0.9× VAHTS DNA Clean Beads. The purified DNAs were amplified through PCR for 11 (pre-enrichment), 12 (labeled and post-enrichment) or 16 (omitting probe and post-enrichment) cycles, and then purified with 0.9× VAHTS DNA Clean Beads. The quality of libraries was assessed by

Fragment Analyzer. Two rounds of size selection were performed with 0.6-0.7× and 0.3× DNA Clean Beads according to the instructions of NEBNext Ultra II RNA Library Prep Kit for Illumina and VAHTS DNA Clean Beads. The final cDNA libraries were sequenced of 150 bp paired-end with ~40 M reads on Illumina HiSeq X Ten platform.

## smFISH validation

Primary probes for targeted mRNAs were designed by Oligostan[52] following the supplemented protocol with FLAP X complementary sequence conjugated at the 3'-end of each probe (see Supplementary Data 4). Primary probes targeted to a mRNA of interest were dissolved to 100 μM in TE buffer (Sigma, 93283), mixed at equal volume, and then diluted in TE buffer to a final concentration of 20 μM. Secondary probes conjugated with Alexa Fluor 647 (Invitrogen, A-21245) was dissolved at 10 μM by TE buffer.

HEK293T and U-2 OS cells stably expressed G3BP1-EGFP fusion protein were cultured to ~50% confluency on glass coverslips pre-coated with matrigel in 24-well plates. Following aforementioned stress treatment, cells were washed once with PBSM (PBS with 1 mM MgCl₂), fixed with paraformaldehyde (3.2% (w/v) in PBSM) for 10 min at room temperature, and washed with cold PBSM in the presence of 10 mM glycine. Thereafter, cells were washed once with PBS and permeabilized on ice for 20 min in PBSM containing 0.1% Triton X-100 and 2 mM vanadyl ribonucleoside complex (VRC, NEB, S1402S). After washed by PBSM, cells were then incubated at room temperature for 15 min by prehybridization-30 buffer consisting of 30% formamide (Sigma, F9037) in 2× SSC (Invitrogen, AM9770). Cells were stained overnight at 37 °C with hybridization buffer containing 10% (v/v) dextran sulfate (Sigma, D6001), 30% formamide, 2× SSC, 2 mM VRC, 10 μg/ml Salmon Sperm DNA Solution (Invitrogen, 15632011), 10 μg/ml E. coli tRNA (Roche, 10109541001), 10 μg/mL BSA (0.22 μm filtered) and 200 ng primary probe mix.

The next day, cells were washed twice with prehybridization-30 buffer, each time incubating at 37 °C for 20 min. Cells were post-fixed in 1% paraformaldehyde in PBSM for 5 min at room temperature, followed by washing twice in 2× SSC and incubation in prehybybridization-10 (10% formamide, 2× SSC) for 10 min at 37 °C. Cells were stained with hybridization buffer containing 10% (v/v) dextran sulfate, 10% formamide, 2× SSC, 2 mM VRC, 10 μg/ml Salmon Sperm DNA Solution, 10 μg/ml E. coli tRNA, 10 μg/ml BSA and 10 pM secondary probe for 3 h at 37 °C. Secondary probes containing Alexa Fluor 647 conjugated at the 3'- and 5'- termini of the FLAP X sequence (see Supplementary Data 4) were synthesized by Invitrogen and stored in Tris-EDTA buffer as 100 μM stock solution. Stained cells were washed twice by prehybybridization-10, each time incubating at 37 °C for 10 min, and washed by 2× SSC. Finally, cells were counterstained with DAPI (diluted in 2× SSC) for 10 min at room temperature. The coverslips with stained cells were mounted on a glass slide in the presence of Fluoromount-G Anti-Fade (SouthernBiotech, 0100-35).

## Fluorescence microscopy

Immunofluorescence imaging was performed on an inverted fluorescence microscope (Nikon-TiE) equipped with a spinning disk confocal unit (Yokogawa CSU-X1) and a scientific CMOS camera (Hamamatsu ORCA-Flash 4.0 v.2). The microscope, camera and laser lines were controlled with a customized software written in LabVIEW v.15.0 (National Instruments). A 40× oil immersion lens was used to acquire immunofluorescence images. All images shown were one Z-plane. The degree of G3BP1-miniSOG and untargeted-miniSOG co-localized to SG were quantified by a custom-made MATLAB script (see Code Availability).

For smFISH imaging, a 60× oil immersion lens with 1.5× magnifying was used to acquire confocal image stacks, with a step size of 0.4 μm along the z-axis. The scan range was set as 6 μm for HEK293T cells and 4 μm for U-2 OS cells. The script for quantitation of

smFISH images was written in MATLAB (R2021a, V9.10.0.1602886). Briefly, the cell boundary was manually outlined from the maximum intensity projection of the G3BP1-EGFP image stack. G3BP1-positive regions were identified as those pixels with intensities higher than a manually determined threshold in the maximum intensity projection image of the G3BP1-EGFP channel. The RNA puncta were identified from the FISH images by selecting those pixels with intensities at least 2-fold higher than the average intensity. RNA puncta less than 12 or 15 pixels (corresponding to 0.063 or 0.078 μm$^2$, depending on the target mRNA) were discarded from further analysis. The extent of RNA colocalization with SG was quantified as the ratio between the number of RNA puncta falling within G3BP1-positive regions and the total number of RNA puncta for a given cell. We also applied the following formula to calculate the level of RNA enrichment in SG:

$$\text{Level of enrichment} = \frac{\text{pixels(RNAs inside SG)/SG area}}{\text{pixels(RNAs outside SG)/(whole cell} - \text{SG area)}}$$

### Sequencing data analysis

The adapters sequence in reads were removed by Cutadapt (v.1.18)[53] and quality controlled by FastQC (v0.11.8) to ensure the adapters were completely removed. Then reads were mapped by hisat2 (v2.1.0)[54] to the human genome assembly GRCh38 (hg38) with gene annotation (v.87) downloaded from Ensembl website. The mapped reads were counted by htseq-count (v0.7.2)[55] with the option "–stranded no".

A series of differential analysis were performed by R package DESeq2 (v1.34.0)[35] to define the SG datasets captured by CAP-seq. The genes with log$_2$(Fold Change) > 0.3 and log$_2$(Fold Change) < −0.3 with adjusted $p$ value ($p_{adj}$) < 0.05 were defined as enriched and depleted respectively in each DESeq2 analysis. The SG-proximial or G3BP1-proxmial RNAs in the aforementioned stress treated cells were defined as the overlap between the enriched targets in three differential analysis: 1) post- versus pre-enrichment of RNA labeled with G3BP1-miniSOG; 2) post-enriched RNA labeled with G3BP1-miniSOG versus RNA from negative control omitting PA; and 3) post-enriched RNA labeled with G3BP1-miniSOG versus untargeted miniSOG under same treatment. The SG/G3BP1-excluded RNAs were defined as the overlap between the depleted targets in 1) and 3) as mentioned above. Four biological replicates of HEK293T cells stably expressing untargeted miniSOG of each treatment were used in analysis. HEK293T cells stably expressing G3BP1-miniSOG fusion protein that recovered for 1 h from arsenite stress were prepared with three replicates. Other samples were prepared with two biological replicates.

### RNA feature analysis

Translational efficiency of each mRNAs were calculated from a previous report (log$_2$(Ribosome protected fragment reads/RNA-seq reads))[56]. m$^6$A sites in HEK283T were obtained from a published report[41]. mRNAs in each dataset were used for feature analysis. Transcript length and GC content in gene were downloaded from Ensembl biomart tool. For CAP-seq captured datasets in U-2 OS, GC content and transcript length from the longest isoform of each transcript were downloaded from Ensembl release 87. For CAP-seq captured datasets in HEK293T, total transcript length of the longest isoform were acquired from Ensembl release 107. Length of the longest 3' UTR, CDS, 5' UTR and AU content within different regions were calculated from the sequence downloaded from Ensembl release 107 based on the stable Ensembl gene ID in CAP-seq datasets. Heatmap was generated by R script pheatmap (v1.0.12) with clustering method as "complete", and the value of log$_2$(Fold Change) counted in DESeq2 analysis between enriched RNAs labeled by G3BP1-miniSOG and untargeted miniSOG were used to scale. Mann-Whitney test was used in statistical analysis by OriginPro (2019).

### Statistics and reproducibility

All western blots have been repeated three times and similar results were obtained. These data are presented in the supplementary information. Quantitative analysis of western blots is shown in bar charts with error bars representing mean ± SD generated from three biological replicates. All immunofluorescence and smFISH images are representative examples from at least three fields of view acquired at three independent biological replicates.

HEK293T cells stably expressing G3BP1-miniSOG under basal, arsenite stress, sorbitol stress or 3 h post-arsentie stress were prepared in two biological replicates for sequencing; HEK293T cells stably expressing G3BP1-miniSOG under 1 h post-arsenite stress were prepared in three biological replicates for sequencing; HEK293T cells stably expressing untargeted miniSOG were prepared in four replicates. U-2 OS cells stably expressing untargeted miniSOG and G3BP1-miniSOG were prepared in two replicates. Western blot and imaging experiments were acquired for at least three independent replicates. The replicates of each experiment showed good reproducibility. All of the raw data for NGS sequencing results could be found in the Data Availability section.

### Reporting summary

Further information on research design is available in the Nature Portfolio Reporting Summary linked to this article.

## Data availability

The data supporting the findings of this study are available from the corresponding authors upon request. The sequencing data reported in this paper have been deposited in the National Center for Biotechnology Information Gene Expression Omnibus (accession code: GSE223295). Western blot data generated in this study are provided in the Source Data file. The raw data of all the bar charts and the associated $p$ values are provided in the Source Data file. Source data are provided with this paper.

## Code availability

The script for analyzing smFISH images and SG co-localization have been deposited on GitHub website: https://github.com/PKUCHEMZouLab/CAP-seq_stress-granule/.

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

## Acknowledgements

This work was supported by the Ministry of Science and Technology (2022YFA1304700, 2018YFA0507600), the National Natural Science

Foundation of China (32088101). P.Z. is sponsored by Bayer Investigator Award.

## Author contributions

P.Z. conceived the project. Z.R., W.T., and P.Z. designed experiments. Z.R., W.T., and L.P. performed experiments. Z.R. and P.Z. analyzed data and wrote the paper with inputs from other authors.

## Competing interests

The authors declare no competing interests.
