## [Peer Review File · Nature Communications]

Profiling stress-triggered RNA condensation with photocatalytic proximity labelingReviewer #1 (Remarks to the Author):

Stress granule (SG) is known for buffering proteomic stress, and its defective function is linked to various neuronal disorders. Identifying the definitive composition of SG is crucial for understanding its related physiology and developing therapy for SG-related diseases. Since SG is considered a liquid-liquid condensate, proximity labeling is the preferred method for identifying its contents because conventional purification methods cannot perfectly isolate it without contaminants. Apex, BioID, and TurboID have nicely revealed the protein components of SG. SG also contains various RNA molecules, and it is believed that the specific RNA and protein interaction supports the formation of SG. Therefore, RNA components in SG are of great interest, and various studies have identified this RNA information with fractionation and Co-IP methods. This method has revealed that RNA molecules can be dynamically trapped in SG under various stress. Recently, Dr. Peng Zou's group developed an efficient proximal RNA labeling method such as APEX-Aniline labeling (Zhou et al. *Angew Chem Int Ed*, 2019) and Cap-seq (Wang et al. *Nat Chem. Biol.* 2019) to identify SG-localized RNA information, which has lower labeling efficiency than that of proteins. The proximal RNA labeling method, developed by Zou group, is also utilized in the recent work of cell surface-localized RNA mapping by Bertozzi's group (Flynn et al. *Cell*, 2021). Thus, I believe that this work of identifying SG-localized RNA using Cap-seq by the Zou group will be a valuable resource that can be extensively exploited in the SG biology community after further comparison with previous findings. After these corrections and additions during the revision process, I recommend publishing this work in Nature Communications.

Major points:

1. There are several important points that the authors need to address in their study. Firstly, as there are previously published studies on SG-enriched RNA using various techniques, the authors need to compare their data with these previous findings. For example, previous approaches did not mention enrichment of AU-rich RNA. If these RNA species are not temporal residents of SG, their absence in previous studies raises questions about whether the molecular information provided by these species is reliable.
2. Secondly, the authors should extensively compare their Cap-seq data with previous SG-RNA profiling studies, such as APEX-seq data (PMID: 31442426). While the labeling mechanism (conjugation to exposed guanosine) is similar to each other, there may be similarities and differences between the two datasets that the authors need to identify.
3. Thirdly, many SG biologists believe that SG proteome is highly cell-type specific. For example, TDP-43, a well-known SG component protein in neurons, is not found in SG proteomics in Hela cells (PMID: 29395067). The authors should investigate whether the current Cap-seq findings of resident RNA molecules such as AU-rich RNA can be considered general components in other cell lines or cell-type specific findings in Hek293 cells. If the authors conduct additional Cap-seq experiments with other cell lines, they can provide more insight into the generalism of their findings. Utilizing neural-originated cell lines for another round of Cap-seq experiment can make this work more valuable as many known SG-

related diseases are neurological disorders, such as ALS.

Minor points:

1. The authors should clarify why they used 10mM PA and whether the increased concentration is required due to the Core-Shell structure of SG, which might hinder probe penetration. The authors should also measure or mention any toxicity or stress caused by probe treatment in the Discussion section.
2. Cu-click protocol used in the immunofluorescence and RNA isolation experiments appear to be slightly different (e.g., copper conc. THPTA vs BTAA), and the authors should provide a brief comment explaining this experimental difference.
3. The authors need to address the limitation of their analysis method, specifically the possibility of indirect identification of RNA. If this indirect issue is present in this study, it needs to be clearly mentioned in the Discussion section.

Reviewer #2 (Remarks to the Author):

The manuscript, "Profiling stress-triggered RNA condensation with photocatalytic proximity labeling" by Ren et al., contributes a novel approach to profile the transcripts that are enriched in stress granules called CAP-seq. The CAP-seq approach employs a miniSOG fusion protein to label and subsequently purify and identify the RNAs that are in proximity to G3BP1-miniSOG in stressed and unstressed conditions. The study demonstrates that there are similarities and differences in the RNAs that are differentially enriched in stress granules depending on cell type, stress type, and time over a stress time-course experiment. The primary novelty of this study is in defining the G3BP1-associated transcriptome during the recovery from stress, which is an important contribution to the field. However, major weaknesses in the data presented and the interpretation of the data must be addressed for this manuscript to be considered for publication. For this reason, I do not recommend publication of this manuscript in Nature Communications. Specific areas to be addressed are listed below.

Major comments:

1. The observations that RNA length, AU rich elements, and m6A elements are associated with RNA enrichment in stress granules are not novel and have already been shown by several groups and published in articles including Khong et al., Mol Cell 2017; Namkoong et al., Mol Cell 2018; Van Treeck et al., PNAS 2018; Ries et al., 2019 Nature; and Matheny et al., Mol Cell Biol 2019. Therefore, these observations should be used to bolster the validity of the novel CAP-seq approach rather than portrayed as novel findings throughout the manuscript.
2. Several key manuscripts are not discussed or included in citations in the introduction of the

manuscript. These include sources listed in comment 1, in addition to the study by Khong et al., Nat Commun 2022 which describes a limited effect of m6A on mRNA recruitment to stress granules. Additionally, Moon et al., Nat Cell Biol 2019 demonstrated that translating mRNAs can transiently colocalize with stress granules, which should be included with the Mateju et al., 2020 Cell citation.

3. G3BP1 is an RNA binding protein, therefore it is expected that mRNAs that bind to G3BP1 will be present in the unstressed and stressed G3BP1-miniSOG datasets. The statement, “It appears that a sub-population of mRNAs remain proximal to G3BP1 in the cytoplasm despite apparent SG disassembly” implies that these detected RNAs are within stress granule cores, but no experimental evidence is presented to support that statement.

4. The m6A enrichment data shown in Figure 4H and 5G must be normalized to total transcript length to be meaningful.

5. The details of the CAP-seq approach must be clarified to indicate the limitations of the assay. The following should be addressed to give confidence that the method is resulting in specific labeling of RNAs in stress granules:

- a. The distance between the miniSOG protein and the RNA target that allows labeling should be stated.
- b. The degree of G3BP1-miniSOG localization to stress granules and the degree of the control miniSOG in stress granules must be quantified and reported, especially considering that miniSOG is present in stress granules.
- c. The definition of ‘post-versus pre-enrichment of RNA’ should be given. Is this equivalent to an ‘input’ sample in an RNA-protein co-immunoprecipitation experiment?
- d. The rationale for pooling the three negative controls (post-versus pre-enrichment, PA-omitted, and untargeted miniSOG) should be clarified and results should be shown for each individual treatment as they are controlling for different aspects of the approach.
- e. It is unclear whether this same approach of pooling all three negative control datasets is used for the unstressed HEK293T cell and the recovery from stress datasets.

6. Poor data quality throughout the manuscript makes the study results difficult to interpret. In particular:

- a. Unstressed cells must be shown in Figure 1B, Figure 2A, Figure 3A, Figure S2,
- b. Statistical analyses must be done for all Venn diagrams to enable a meaningful interpretation to be made (Figure 1D, Figure 2C, Figure 2D, Figure 3C, Figure 3D, Figure S5, Figure S12, Figure S16; Figure S19, Figure S20)
- c. Loading controls and results from experimental replicates must be included for all western blots (Figure S1, Figure S6, Figure S18)
- d. Some smFISH images do not appear to have detectable RNA and/or stress granules in the cytoplasm, making them uninterpretable (Figure 1E CCNL2, Figure 2E SRRM2, Figure S8 PKD1)
- e. The results of independent experimental replicates for the smFISH data (e.g., Figure 1F) should be reported rather than the number of individual cells quantified.

7. The stress granule transcriptome generated by Khong et al. 2017 Mol Cell is treated throughout the manuscript as if all the RNAs within it are enriched in stress granules, however, this dataset contains all detectable RNAs that are in stress granules at varying degrees. This is an important distinction because CAP-seq results throughout the paper are not generally showing different results from previously published data in Khong et al., 2017.

8. Several statements in the text are misrepresenting the presented data:

The statement “No significant differences in TE and transcript length were found among mRNAs in stress-independent, sorbitol-specific and arsenite-specific datasets” is not supported by the data shown in Figure S14, which shows significant differences in TE and transcript length in the arsenite-specific dataset.

The statement “At 3 hr post-stress, the eIF2a phosphorylation appears fully restored to its basal level” is not supported by the data in Figure S18 which shows P-eIF2 is 2-3x higher at 3 hr compared to the basal unstressed condition.

Minor comments:

References to Parker et al., and Jeffrey (Jaffrey) et al., should be replaced with the citation in proper format.

The observation that mitochondrial gene encoded RNAs are depleted from stress granules using CAP-seq is not novel as it is stated in Khong et al., Mol Cell 2017 that mitochondrial RNAs present in their transcriptome dataset may be contaminants introduced during purification.

The source of the data used to create translation efficiency and transcript length plots throughout should be defined in the figure legends.

Reviewer #3 (Remarks to the Author):

In this manuscript, Ren and colleagues use the RNA proximity labeling approach CAP-seq to identify and quantify transcripts localized to stress granules (SG). They use this technique to profile how the SG transcriptome changes in response to different stresses (arsenite vs. sorbitol) and how its RNA content dynamically changes upon SG disassembly. Generally, the conclusions in the manuscript are supported by the data. I have only a few comments that may improve the manuscript.

MAJOR COMMENTS

1. Again and again, the authors find that SG-proximal RNAs are (1) long, (2) AU-rich, and (3) poorly

translated. They ascribe function to all of these characteristics and note that they may be involved in localizing RNAs to SG. However, across the transcriptome, all 3 of these characteristics are correlated with each other. That is, RNAs that are long also tend to be AU-rich, and vice versa (Marin et al Yeast 2003; Lopez et al Frontiers in Genetics 2021, and others). It could be then that just one of these characteristics is important and the others come along for the ride through these correlations but are not actually functional themselves in terms of getting RNAs to SGs. This point should at least be noted.

2. Related to point 1, I particularly disagree with the statement that “RNA binding proteins generally prefer AU-rich motifs”. Yes, many RBPs do bind AU-rich sequences, but many also do not. Characterizing the entirety of the RNA binding proteome in this way is not accurate.

3. Any time that membership in a group is tested across experiments (e.g. Venn diagrams comparing SG-enriched RNAs across stresses or cell types, etc.), p-values should be calculated for the overlap. Without some kind of test, it’s difficult for me to know whether or not the overlap observed is more or less than expected. This is important as observing a greater overlap than expected would lend additional confidence to the results.

MINOR COMMENTS

1. Where did the translational efficiency data come from? I may have missed it, but it was not obvious to me. This should be made clearer.

2. “Transcription factors binding to AU-rich elements [have] been found in SG.” Are you sure you mean transcription factors? Those don’t normally bind RNA.

3. “However, the purification procedure is prone to contamination and loss of weakly associated material, thus causing high false positive rate.” The loss of material that in truth really is associated with SG would result in false negatives, not false positives.

We thank all three reviewers for their thorough and thoughtful comments to help us improve this manuscript. In the revised manuscript, we have provided additional experimental data on the evaluation of miniSOG expression levels and labeling-induced toxicity. We have also provided additional statistical analysis of the co-localization between G3BP1-miniSOG and stress granules and compared our CAP-seq datasets between HEK293T and U-2 OS cells to highlight cell-type specific RNA enrichment. As suggested by the reviewers, we have added discussions to compare our findings with previous reports, especially regarding the RNA features associated with SG enrichment. Please see our point-by-point responses below. In the revised manuscript, these changes have been marked in red.

Reviewer #1 (Remarks to the Author):

Stress granule (SG) is known for buffering proteomic stress, and its defective function is linked to various neuronal disorders. Identifying the definitive composition of SG is crucial for understanding its related physiology and developing therapy for SG-related diseases. Since SG is considered a liquid-liquid condensate, proximity labeling is the preferred method for identifying its contents because conventional purification methods cannot perfectly isolate it without contaminants. Apex, BioID, and TurboID have nicely revealed the protein components of SG. SG also contains various RNA molecules, and it is believed that the specific RNA and protein interaction supports the formation of SG. Therefore, RNA components in SG are of great interest, and various studies have identified this RNA information with fractionation and Co-IP methods. This method has revealed that RNA molecules can be dynamically trapped in SG under various stress. Recently, Dr. Peng Zou's group developed an efficient proximal RNA labeling method such as APEX-Aniline labeling (Zhou et al. *Angew Chem Int Ed*, 2019) and Cap-seq (Wang et al. *Nat Chem. Biol.* 2019) to identify SG-localized RNA information, which has lower labeling efficiency than that of proteins. The proximal RNA labeling method, developed by Zou group, is also utilized in the recent work of cell surface-localized RNA mapping by Bertozzi's group (Flynn et al. *Cell*, 2021). Thus, I believe that this work of identifying SG-localized RNA using Cap-seq by the Zou group will be a valuable resource that can be extensively exploited in the SG biology community after further comparison with previous findings. After these corrections and additions during the revision process, I recommend publishing this work in *Nature Communications*.

Major points:

1. There are several important points that the authors need to address in their study. Firstly, as there are previously published studies on SG-enriched RNA using various techniques, the authors need to compare their data with these previous findings. For example, previous approaches did not mention enrichment of AU-rich RNA. If these RNA species are not temporal residents of SG, their absence in previous studies raises questions about whether the molecular information provided by these species.

Response: We thank the reviewer for the advice on comparing our data with previous findings regarding the features of SG transcriptome. In the previous submission, we only compared our CAP-seq dataset with the SG_{coreRNA} dataset derived from arsenite stressed U-2OS (by Parker and co-workers). In the revised manuscript, we have provided additional comparisons with several other published SG-enriched RNA datasets, as suggested by both reviewers #1 and #2.

In the work by Namkoong *et al.* and co-workers (PMID: 29576526), RNA granules were purified from NIH-3T3 cells stressed with thapsigargin, heat shock, or sodium arsenite. The authors found that these stress-induced RNA granules tend to enrich transcripts with longer length and more AU-elements. Notably, 181 out of 457 transcripts enriched in our CAP-seq dataset are also enriched in their arsenite treated NIH-3T3 dataset (Clusters 1 and 2 in Figure 7C in their paper, 2920 transcripts in total), and both our studies identified the same trend of length distributions in SG-enriched RNAs. However, the purification-based methods may have lower SG-specificity, as the purified RNA granules might contain components from other cytosolic membrane-less organelle such as processing bodies.

The correlation between RNA length and SG-enrichment have also been investigated by Van Treeck *et al.* (PMID: 29483269) and Matheny *et al.* (PMID: 31591142). Van Treeck *et al.* analyzed the components of liquid-liquid phase separation (LLPS) droplets formed by yeast RNAs *in vitro*. They found similar RNA components in these droplets as those in the SGs in yeast, and droplet-enriched RNAs contained longer transcripts than droplet-depleted RNAs. Matheny *et al.* analyzed the RNA components of purified RNA granules from arsenite-treated U-2 OS cells, where they found impaired translation efficiency and longer transcript length.

More recently, Kleiner and co-workers applied proximity-dependent RNA editing technique (TRIBE-ID) to profile G3BP1-associated RNAs in HEK293T cells challenged with sodium arsenite stress (PMID: 37349582). Their data revealed that G3BP1-associated RNAs are positively correlated with their transcript length and negatively correlated with translation efficiency, which is consistent with our findings. Among the 745 G3BP1-associated RNAs identified in their dataset, 106 were also enriched in our CAP-seq SG RNA datasets derived from arsenite stressed HEK293T.

All of the above references have been added to the revised manuscripts:

Page 9, lines 12 - 13: Previous studies have revealed a negative correlation between mRNA association with SG and translational efficiency (TE) (NamKoong, *et al.*, *Mol. Cell*, 2017; Khong, *et al.*, *Mol. Cell*, 2017; Padron, *et al.*, *Mol. Cell*, 2019; Matheny, *et al.*, *Mol. Cell Biol.*, 2019)

Page 9, lines 15 - 19: We also observed longer transcript length and higher AU content in SG-proximal versus SG-excluded mRNAs (Figure S9, Table S2B), which follows a similar trend as previous reports that SG_{coreRNA} contained less GC content. (Khong, *et al.*, *Mol. Cell*, 2017). The above analysis shows that data acquired with CAP-seq are generally in agreement with prior knowledge of SG transcriptome.

Page 19, line 24 - page 20, line 8: “For example, mRNAs *SRRM2*, *PLXNB2*, *PKD1* and *MT-ND4*, were previously enriched from purified SG (Khong, *et al.*, *Mol. Cell*, 2017) but were found to be SG-excluded by CAP-seq. Our quantitative smFISH imaging analysis shows less than 10% co-localization of these RNAs with SG, thus supporting the CAP-seq dataset. We also compared our SG-proximal datasets with RNAs enriched in granules isolated from arsenite-stressed NIH-3T3 cells (NamKoong, *et al.*, *Mol. Cell*, 2017), revealing an overlap of 181 transcripts. Notably, both studies identified a positive correlation between transcript length and SG-enrichment. Such correlation has also been reported by previous research on liquid-liquid phase separation of yeast RNAs *in vitro* (Van Treeck, *et al.*, *Proc. Natl. Acad. Sci. U.S.A.*, 2018) and RNA granules purified from arsenite-stressed U-2 OS cells (Matheny, *et al.*, *Mol. Cell Biol.*, 2019).”

Page 20, lines 19 - 25: “More recently, Kleiner and co-workers applied proximity-dependent RNA editing technique (TRIBE-ID) to profile G3BP1-associated RNAs in HEK293T cells challenged with sodium arsenite stress (Seo & Kleiner, *Nat. Chem. Biol.*, 2023). Among the 745 G3BP1-associated RNAs identified in their dataset, 106 were enriched in our CAP-seq SG RNA datasets derived from arsenite stressed HEK293T (p value $< 4 \times 10^{-100}$). Notably, their data revealed that G3BP1-associated RNAs are positively correlated with their transcript length and negatively correlated with translation efficiency, which is consistent with our findings.”

Regarding our statement on AU-rich RNAs, it is noteworthy that our definition of AU content is essentially the opposite to the definition of GC content in previously published reports, i.e. $AU\% = 100\% - GC\%$. Thus, AU-rich means low GC content. The correlation between SG enrichment and GC content has been reported in previous studies. For example, Khong *et al.* (PMID: 29129640) purified the SG cores from arsenite-stressed U-2OS cells and found less GC content in the RNA components. Therefore, our finding of AU-rich RNAs in SG-proximal transcriptome does not contradict with previous reports. We have clarified this point in the Methods section. In

the revised manuscript, we have added a comment on AU-rich and GC-poor in the main text:

Page 9, lines 15 - 18: We also observed longer transcript length and higher AU content in SG-proximal versus SG-excluded mRNAs (Figure S9, Table S2B), which follows a similar trend as previous reports that SG_{coreRNA} contained less GC content. (Khong, *et al.*, *Mol. Cell*, 2017)

2. Secondly, the authors should extensively compare their Cap-seq data with previous SG-RNA profiling studies, such as APEX-seq data (PMID: 31442426). While the labeling mechanism (conjugation to exposed guanosine) is similar to each other, there may be similarities and differences between the two datasets that the authors need to identify.

Response: We agree with the reviewer that it would be informative to compare our CAP-seq dataset with APEX-seq, especially when considering their similar labeling mechanisms. In the work by Padron, *et al.* (PMID 31442426), as recommended by the reviewer), the translation factor eIF4A was used as the bait to fuse with APEX2, and SG formation was induced by heat-shock in HEK293T cells. Unfortunately, the authors did not provide a list of enriched SG-proximal transcripts in the published paper, so we started with re-analyzing their raw RNA-seq data of enriched and input samples. By applying the same cutoff as in our analysis ($\log_2FC > 0,3$ and $p_{adj} < 0.05$), differential expression analysis of the APEX-seq dataset reveals 394 transcripts as significant enriched. Among these, only 18 transcripts overlap with our CAP-seq dataset of 457 transcripts. The lack of overlap may be attributed to the differences in our choices of baits (G3BP1 vs. eIF4A), stress conditions (sodium arsenite vs. heat shock), and duration of stress (1 hr vs. 20 min). Nevertheless, both our current study and the APEX-seq study identified a strong correlation between SG-proximal RNA labeling and transcript length, suggesting that longer RNAs are more likely localized to the SG. In the revised manuscript, we have added the above comparison and discussions:

Page 20, lines 9 - 18: "We also compared our CAP-seq dataset with a previously reported SG transcriptome profiling with APEX-seq, in which the translation factor eIF4A was used as the bait and SG formation was induced by heat-shock for 20 min in HEK293T cells (Padron, *et al.*, *Mol. Cell*, 2019). By applying the same cutoff as in our analysis ($\log_2FC > 0.3$ and $p_{adj} < 0.05$), differential expression analysis of the APEX-seq dataset reveals 394 transcripts as significant enriched. Among these, only 18 transcripts overlap with our CAP-seq dataset of 457 transcripts, which may be attributed to the differences in the choices of baits, stress conditions, and durations of stress. Nevertheless, both datasets identified a strong correlation between SG-proximal RNA labeling and transcript length, suggesting that longer RNAs are more

likely localized to the SG.”

3. Thirdly, many SG biologists believe that SG proteome is highly cell-type specific. For example, TDP-43, a well-known SG component protein in neurons, is not found in SG proteomics in Hela cells (PMID: 29395067). The authors should investigate whether the current Cap-seq findings of resident RNA molecules such as AU-rich RNA can be considered general components in other cell lines or cell-type specific findings in Hek293 cells. If the authors conduct additional Cap-seq experiments with other cell lines, they can provide more insight into the generalism of their findings. Utilizing neural-originated cell lines for another round of Cap-seq experiment can make this work more valuable as many known SG-related diseases are neurological disorders, such as ALS.

Response: We thank the reviewer for the advice on comparing CAP-seq findings in different cell lines. As the reviewer pointed out, SG proteome is highly cell-type specific. Previous proteomic profiling experiments identified 123 and 411 SG proteins in arsenite-stressed HEK293T (Markmiller *et al.*, *Cell*, 2018; PMID: 29373831) and U-2 OS cells (Jain *et al.*, *Cell*, 2016; PMID: 26777405), respectively. Among these, only 82 proteins were enriched in both datasets. It is therefore expected that the SG transcriptome might also be cell-type specific. In our previous submission, we have performed CAP-seq in both HEK293T and U-2 OS cells but did not compare these datasets. In the revised manuscript, we have provided additional data analysis by comparing these datasets in a head-to-head manner. Such comparisons are particularly meaningful as they employ the same labeling method with the same bait protein. Our CAP-seq analysis identified a total of 457 and 1135 transcripts from arsenite-stressed HEK293T and U-2 OS cells, respectively. Among these, 261 transcripts are shared by both datasets (i.e. cell type-independent). The remaining 196 transcripts in the HEK293T SG CAP-seq dataset and 874 transcripts in the U-2OS SG CAP-seq dataset are defined as HEK293T-specific and U-2OS specific, respectively.

We also compared the RNA features of each category. Overall, we did not observe substantial differences in the translation efficiency, AU content, and transcript length. U-2 OS-specific CAP-seq RNAs tend to have higher AU content and shorter CDS length. The trend is quite modest yet statistically significant. These additional results are presented in Figure S13 - S14 in the revised manuscript.

Figure S13. Venn diagram comparing SG-proximal RNAs from arsenite-stressed HEK293T and U-2 OS cells.

Figure S14. Box plot comparing the translation efficiencies (left), AU content (middle) and transcript length (right) between HEK-specific, cell type-independent and U-2 OS-specific mRNAs. ns, not significant ($p > 0.05$); * $p < 0.05$; ** $p < 0.01$; *** $p < 0.001$; **** $p < 0.0001$ (Mann-Whitney test). Translation efficiencies are counted from a previous study (Sidrauski, *et al.*, *Elife*, 2015). AU content and transcript length features are referenced from Ensembl website.

We thus conclude that while the SG RNA compositions may vary substantially in different cell lines, presumably due to differences in the SG protein components that are responsible for recruiting these RNAs, the overall SG RNA features are quite similar, suggesting a conserved mechanism of RNA sorting into the SG. In the revised manuscript, we have added the following comparisons

Page 9, line 21 - page 10, line 5: “Notably, the molecular composition of SG is known to be cell-type specific. For example, TDP-43, the well-known SG protein marker in neuron, is not found in the SG proteome in Hela cells. In addition, proteomic profiling experiments identified 123 and 411 SG proteins in arsenite-stressed HEK293T and U-2 OS cells, by APEX proximity labeling or purification of SG, respectively. Among these, only 82 proteins were enriched in both datasets. It is therefore expected that the SG transcriptome might also be cell-type specific. We compared the 457 and 1135 SG-proximal transcripts from arsenite-stressed HEK293T and U-2 OS cells, respectively. Among these, 261 transcripts are shared by both datasets (i.e. cell type-independent). The remaining 196 transcripts in the HEK293T

SG CAP-seq dataset and 874 transcripts in the U-2 OS SG CAP-seq dataset are defined as HEK293T-specific and U-2 OS-specific, respectively (Figure S13).

We compared the RNA features of cell type-specific and cell type-independent categories, but did not observe substantial differences in the translation efficiency, AU content, and transcript length. U-2 OS-specific CAP-seq RNAs tend to have higher AU content and shorter CDS length (Figure S14). The trend is quite modest yet statistically significant. The above comparisons indicate that while the SG RNA compositions may vary substantially across different cell lines, presumably due to differences in the SG protein components that are responsible for recruiting these RNAs, the overall SG RNA features are quite similar, suggesting a conserved mechanism of RNA sorting into the SG.”

We also thank the reviewer for suggesting CAP-seq experiments in neural-originated cells. Projects of SG CAP-seq in cultured cortical neurons are already underway in our laboratory. Due to difficulties associated with culturing primary cells and labeling in these delicate samples, these projects involve engineering a better version of CAP-seq labeling with directed evolution, which, in our opinion, are beyond the scope of the current study. In the revised manuscript, we have provided additional discussions on the potential limitation of the current CAP-seq method and future directions that aims to improve its efficiency and compatibility with delicate samples such as primary neuron culture.

Minor points:

1. The authors should clarify why they used 10mM PA and whether the increased concentration is required due to the Core-Shell structure of SG, which might hinder probe penetration. The authors should also measure or mention any toxicity or stress caused by probe treatment in the Discussion section.

Response: The choice of 10 mM of PA is based on our previously published CAP-seq method (Wang *et al.* Nat Chem Biol, 2019), which showed that incubating cells with 10 mM PA for 5 min provided sufficient labeling efficiency in multiple subcellular compartments, including endoplasmic reticulum membrane, outer mitochondrial membrane, and mitochondrial matrix. Notably, CAP-seq labeling in the mitochondrial matrix is particularly challenging, as the small molecule probe PA has to rapidly penetrate the barriers of both outer and inner mitochondrial membranes. Thus, we reasoned that incubating cells with 10 mM PA for 5 min should allow the probe to permeate through other subcellular structures, including LLPS condensates. In the

revised manuscript, we have added the following comments to clarify our choice of 10 mM PA and 5 min incubation time:

Page 4 lines 11 - 15: “For RNA labeling, our previous CAP-seq experiments showed that 10 mM PA was capable of permeating throughout the cytoplasm within 5 min (Wang, *et al.*, *Nat. Chem. Biol.*, 2019). Since the core-shell structure of SG might hinder probe penetration, we pre-incubated cells with 10 mM PA for 5 min and subsequently illuminated the sample with 24 mW/cm² blue LED for 15 min (Figure S4).”

The reviewer has raised an important issue regarding the potential cytotoxicity of our method. In the previous submission, we have provided confocal fluorescence images of unstressed G3BP1-miniSOG or untargeted-miniSOG cells. We did not observe any visible cytoplasmic puncta following CAP-seq labeling, suggesting that the toxicity of CAP-seq treatment was quite minimal as compared to sodium arsenite treatment. However, we agree with the reviewer that we should evaluate the cytotoxicity effect more carefully. In the revised manuscript, we used eIF2 α phosphorylation as a marker for cellular stress and measured its levels by Western blotting of whole cell lysate following CAP-seq labeling (i.e. pre-incubating cells with 10 mM PA for 5 min at 37°C, before blue LED illumination at room temperature for 15 min at 24 mW/cm², same as our cellular labeling conditions).

Figure S5. Western blot analysis of eIF2 α phosphorylation (p-eIF2 α) in G3BP1-miniSOG HEK293T cells after CAP-seq labeling. The intensity of p-eIF2 α is normalized with respect to α -Tubulin.

As shown in revised Figure S5, Western blot analysis reveals that CAP-seq labeling causes the level of eIF2 α phosphorylation to increase by ~70% from the basal

level. Treating the cells with blue LED illumination alone or incubating cells with 10 mM PA alone causes the eIF2 α phosphorylation levels to increase by ~25% and ~60%, respectively. We reasoned that, as these increases are modest as compared to arsenite-induced stress (elevating eIF2 α phosphorylation by 200%), the threshold of SG assembly has not been reached. Our data analysis workflow requires comparing the enrichment levels of RNAs labeled in G3BP1-miniSOG cells versus untargeted-miniSOG cells, which offers a means of normalizing the slight cellular toxicity effect, as both cell samples were treated by CAP-seq in the identical manner. The observation of PA-induced toxicity also prompts us to reduce the incubation time in the future via improvement in CAP-seq labeling efficiency, which is underway in our lab through coordinated efforts of probe design/synthesis and directed evolution of the photocatalyst.

In the revised manuscript, we have acknowledged the cytotoxicity issue in the current study, discussed the necessity of using appropriate control samples to normalize the cytotoxicity effect, and future directions that aim to mitigate the toxicity via more efficient photocatalysis.

Page 4, lines 15 - 20: “We measured the level of eIF2 α phosphorylation as a marker for translation inhibition. Western blot analysis reveals that CAP-seq labeling elevates eIF2 α phosphorylation by ~70% from the basal level. Treating cells with blue LED illumination alone or incubation with 10 mM PA alone causes the eIF2 α phosphorylation levels to increase by ~25% and ~60%, respectively (Figure S5). These changes are modest as compared to arsenite-induced stress (Figure S2).

2. Cu-click protocol used in the immunofluorescence and RNA isolation experiments appear to be slightly different (e.g., copper conc. THPTA vs BTTAA), and the authors should provide a brief comment explaining this experimental difference.

Response: Thanks for pointing out this issue. In the revised manuscript, we have provided additional experiment data comparing THPTA and BTTAA ligands for their effect on the CuAAC efficiency in fixed cells. Fluorescence imaging reveals similar levels of biotinylation by the two ligands (see figure below). Our choice of THPTA ligand is based on a previous report that CuAAC with THPTA achieves higher reaction yield than BTTAA in the aqueous solution (PMID: 29636419). The CuAAC protocol used in this study is also the same as our previous work of CAP-seq (PMID: 31591565). In the revised manuscript, we have explained the reason of choosing THPTA as the ligand:

Page 4, lines 26 - 27: “We chose THPTA as the ligand for CuAAC because it offers higher reaction yield with RNA in the aqueous solution (Huang, *et al.*, *Proc. Natl. Acad. Sci. U.S.A.*, 2018).”

Figure for Reviewers: Immunofluorescence microscopy of U-2 OS cells expressing G3BP1-miniSOG (green). Cells were treated with 0.5 mM sodium arsenite for 60 min and labeled by 10 mM PA under 15 min blue light illumination. THPTA and BTAA represents CuAAC reaction with THPTA (100 μ M N₃-PEG₃-biotin, 2 mM THPTA, 0.5 mM CuSO₄ and 5 mM sodium ascorbate) or BTAA (50 μ M N₃-PEG₃-biotin, 2 mM CuSO₄, 1 mM BTAA and 0.5 mg/mL sodium ascorbate), respectively. Biotinylated signals are shown in magenta. Scale bars, 10 μ m.

3. The authors need to address the limitation of their analysis method, specifically the possibility of indirect identification of RNA. If this indirect issue is present in this study, it needs to be clearly mentioned in the Discussion section.

Response: We agree with the reviewer that, for the benefit of the readers and future user of CAP-seq, the limitations of our work should be discussed in more detail. These include the slight cytotoxicity effect, limited temporal resolution, and potential bias towards exposed RNAs. First, as we have discussed in our response to Minor Point #1, prolonged treatment of PA causes slight cellular stress. In HEK293T and U-2 OS cells, this level of stress has not reached the threshold of inducing SG assembly and could be normalized by using appropriate controls. However, the mild toxicity of PA treatment may impede the application of CAP-seq to more sensitive and delicate cells such as primary neuron cultures.

Second, the current version of CAP-seq requires blue LED illumination for 15 min to achieve efficient RNA labeling. This temporal resolution is sufficient for mapping the SG transcriptome during its disassembly stage, which occurs on the time scale of hours. Yet, the low temporal resolution is incompatible for resolving SG components during its rapid assembly stage, which typically matures within 15 – 30 min (Wheeler, *et al.*, *Elife*, 2016).

Third, proximity labeling method is potentially biased towards prey proteins/RNAs that are more solvent accessible, such that the reactive singlet oxygen could reach the exposed targets (tyrosine, lysine, guanosine nucleobase, histidine, etc.) for efficient labeling. Notably, this bias is shared by APEX2, BioID/TurboID, CAP-seq, RinID, etc.

In the revised manuscript, we have added the following discussions:

Page 23, lines 18 - 31: “Finally, this study is limited by mild cytotoxicity, insufficient temporal resolution, and potential bias towards exposed RNAs. First, prolonged treatment of PA causes cellular stress, as revealed by elevated eIF2 α phosphorylation. Although the level of stress has not reached the threshold of inducing SG assembly and could be normalized by using appropriate controls, it might still impede the application of CAP-seq to more sensitive and delicate cells such as primary neuron cultures. Second, CAP-seq labeling is potentially biased towards prey RNAs that are more solvent exposed, such that the reactive singlet oxygen could access guanosine nucleobase for successful labeling. Notably, this bias is shared by other proximity labeling methods, such as APEX2 (Fazal, *et al.*, *Cell*, 2019), BioID (Roux, *et al.*, *J. Cell Biol.*, 2012)/TurboID (Branon, *et al.*, *Nat. Biotechnol.*, 2018), and RinID (Zheng, *et al.*, *Nat. Commun.*, 2023). Finally, CAP-seq requires blue LED illumination for 15 min to achieve efficient RNA labeling. This temporal resolution is still insufficient for resolving SG components during its rapid assembly stage, which typically matures within 15 – 30 min. In the future, more effective tools for photosensitive catalysis could be developed by directional evolution.”

Regarding the issue of indirect identification of RNA, as pointed out by the reviewer, we reason that the labeling radius of CAP-seq is the key. While the exact radius has not been measured in the context of live cells, the half-life and diffusion radius of singlet oxygen were estimated to be 0.6 μ s and 70 nm, respectively (Moan, *J. Photoch. Photobio.*, 1990). Since the half-life of singlet oxygen depends critically on the local environment and considering the quenching reaction from cellular metabolites such as thiols and amines, the CAP-seq labeling is likely quite restricted. In our previous CAP-seq profiling at the surface of endoplasmic reticulum membrane, 96.2% of the captured mRNAs encode for the secretory pathway proteins, thus suggesting a spatial resolution on the scale of the size of a ribosome, i.e. 20 nm. As the size of SGs is in the range of 100 nm – 1 μ m, a resolution on the scale of tens of nanometers is sufficient for mapping SG components.

Nevertheless, the labeling radius of CAP-seq is still much larger than the size of a typical protein (e.g. 3 - 5 nm). Thus, CAP-seq is indeed capable of labeling RNAs that are indirectly interacting with the bait. This scenario is shared with other proximity labeling approaches and is quite different from direct capture methods such as chemical- or photo-crosslinking (e.g. CLIP, PAR-CLIP). In the revised manuscript, we have discussed the differences in methodology.

Page 3, lines 10 - 17: “Due to the limited half-life and diffusion distance of singlet oxygen (0.6 μ s and 70 nm in cells) (Moan, *J. Photoch. Photobio.*, 1990), CAP-seq have been applied in profiling the subcellular transcriptome in the mitochondria and

endoplasmic reticulum with high spatial resolution in live cells. As the size of SGs is in the range of 100 nm – 1 μ m (Wolozin & Ivanov, *Nat. Rev. Neurosci.*, 2019), a resolution on the scale of tens of nanometers is sufficient for mapping SG components. Notably, CAP-seq is capable of labeling RNAs that are indirectly interacting with the bait, which could complement direct capture methods such as chemical- or photo-crosslinking (e.g. CLIP, PAR-CLIP).

Reviewer #2 (Remarks to the Author):

The manuscript, “Profiling stress-triggered RNA condensation with photocatalytic proximity labeling” by Ren et al., contributes a novel approach to profile the transcripts that are enriched in stress granules called CAP-seq. The CAP-seq approach employs a miniSOG fusion protein to label and subsequently purify and identify the RNAs that are in proximity to G3BP1-miniSOG in stressed and unstressed conditions. The study demonstrates that there are similarities and differences in the RNAs that are differentially enriched in stress granules depending on cell type, stress type, and time over a stress time-course experiment. The primary novelty of this study is in defining the G3BP1-associated transcriptome during the recovery from stress, which is an important contribution to the field. However, major weaknesses in the data presented and the interpretation of the data must be addressed for this manuscript to be considered for publication. For this reason, I do not recommend publication of this manuscript in Nature Communications. Specific areas to be addressed are listed below.

Major comments:

1. The observations that RNA length, AU rich elements, and m6A elements are associated with RNA enrichment in stress granules are not novel and have already been shown by several groups and published in articles including Khong et al., *Mol Cell* 2017; Namkoong et al., *Mol Cell* 2018; Van Treeck et al., *PNAS* 2018; Ries et al., 2019 *Nature*; and Matheny et al., *Mol Cell Biol* 2019. Therefore, these observations should be used to bolster the validity of the novel CAP-seq approach rather than portrayed as novel findings throughout the manuscript.

Response: We thank the reviewer for pointing out this issue. Indeed, our observed strong correlations between transcript length, AU content, and m⁶A levels with SG RNA enrichment are overall consistent with previous reports. In the previous submission, we have briefly mentioned this point in the Introduction and provided three references (refs 7-9), but we agree with the reviewer that we should add more references to better

place our findings in the context of existing literature. In the revised manuscript, we have referenced the papers recommended by the reviewer and compared our data with these previous reports, as follows:

In the work by Namkoong *et al.* (PMID: 29576526), RNA granules were purified from NIH-3T3 cells stressed with thapsigargin, heat shock, or sodium arsenite. The authors found that these stress-induced RNA granules tend to enrich transcripts with longer length and more AU-elements. Notably, 181 out of 457 transcripts enriched in our CAP-seq dataset are also enriched in their arsenite treated NIH-3T3 dataset (Clusters 1 and 2 in Figure 7C in their paper, 2920 transcripts in total), and both our studies identified the same trend of length distributions in SG-enriched RNAs. However, the purification-based methods may have lower SG-specificity, as the purified RNA granules might contain components from other cytosolic membrane-less organelle such as processing bodies.

The correlation between RNA length and SG-enrichment have also been investigated by Van Treeck *et al.* (PMID: 29483269) and Matheny *et al.* (PMID: 31591142). Van Treeck *et al.* analyzed the components of liquid-liquid phase separation (LLPS) droplets formed by yeast RNAs *in vitro*. They found similar RNA components in these droplets as those in the SGs in yeast, and droplet-enriched RNAs contained longer transcripts than droplet-depleted RNAs. Matheny *et al.* analyzed the RNA components of purified RNA granules from arsenite-treated U-2 OS cells, where they found impaired translation efficiency and longer transcript length.

Regarding RNA methylations, Ries *et al.* (PMID: 31292544) found that RNA methylation m⁶A could induce LLPS condensate formation between m⁶A-mRNAs and their binding proteins. By comparing their RNA m⁶A dataset with a previously reported list of RNA components derived from purified granules (Namkoong *et al.* PMID: 29576526), they discovered that SG RNAs contain more m⁶A modification.

More recently, Kleiner and co-workers applied proximity-dependent RNA editing technique (TRIBE-ID) to profile G3BP1-associated RNAs in HEK293T cells challenged with sodium arsenite stress. Their data revealed that G3BP1-associated RNAs are positively correlated with their transcript length and negatively correlated with translation efficiency, which is consistent with our findings. Among the 745 G3BP1-associated RNAs identified in their dataset, 106 were also enriched in our CAP-seq SG RNA datasets derived from arsenite stressed HEK293T.

All of the above references have been added to the revised manuscripts.

Page 9, lines 12 - 13: "Previous studies have revealed a negative correlation between mRNA association with SG and translational efficiency (TE) (NamKoong, *et al.*, *Mol. Cell*, 2017; Khong, *et al.*, *Mol. Cell*, 2017; Padron, *et al.*, *Mol. Cell*, 2019; Matheny, *et al.*, *Mol. Cell Biol.*, 2019)"

Page 9, lines 15 -20: “We also observed longer transcript length and higher AU content in SG-proximal versus SG-excluded mRNAs (Figure S9, Table S2B), which follows a similar trend as previous reports that SG_{coreRNA} contained less GC content. (Khong, *et al.*, *Mol. Cell*, 2017). The above analysis shows that data acquired with CAP-seq are generally in agreement with prior knowledge of SG transcriptome.”

Page 19, line 24 - page 20, line 8: “For example, mRNAs *SRRM2*, *PLXNB2*, *PKD1* and *MT-ND4*, were previously enriched from purified SG (Khong, *et al.*, *Mol. Cell*, 2017) but were found to be SG-excluded by CAP-seq. Our quantitative smFISH imaging analysis shows less than 10% co-localization of these RNAs with SG, thus supporting the CAP-seq dataset. We also compared our SG-proximal datasets with RNAs enriched in granules isolated from arsenite-stressed NIH-3T3 cells (NamKoong, *et al.*, *Mol. Cell*, 2017), revealing an overlap of 181 transcripts. Notably, both studies identified a positive correlation between transcript length and SG-enrichment. Such correlation has also been reported by previous research on liquid-liquid phase separation of yeast RNAs *in vitro* (Van Treeck, *et al.*, *Proc. Natl. Acad. Sci. U.S.A.*, 2018) and RNA granules purified from arsenite-stressed U-2 OS cells (Matheny, *et al.*, *Mol. Cell Biol.*, 2019).”

2. Several key manuscripts are not discussed or included in citations in the introduction of the manuscript. These include sources listed in comment 1, in addition to the study by Khong *et al.*, *Nat Commun* 2022 which describes a limited effect of m⁶A on mRNA recruitment to stress granules. Additionally, Moon *et al.*, *Nat Cell Biol* 2019 demonstrated that translating mRNAs can transiently colocalize with stress granules, which should be included with the Mateju *et al.*, 2020 *Cell* citation.

Response: We agree with the reviewer that these key references should be included in the manuscript. In the work by Khong *et al.*, an RNA m⁶A methylation-deficient mouse embryonic stem cell line was created to compare the level of SG partitioning between m⁶A-modified RNAs and m⁶A-deficient RNAs. Fluorescence microscopy analysis reveals a similar level of SG enrichment in RNA methylation-deficient cells and wild-type cells, leading the authors to conclude that transcript lengths, rather than m⁶A levels, are correlated with SG enrichment. In the revised manuscript, we have cited this reference in the Introduction and added the following discussions:

Page 2, lines 12 - 17: “For example, m⁶A modification is found to be enriched in granules isolated from arsenite-stressed NIH-3T3 cells (NamKoong, *et al.*, *Mol. Cell*, 2017; Ries, *Nature*, 2019). However, fluorescence microscopy analysis reveals a similar level of SG enrichment in RNA methylation-deficient mouse embryonic stem cells versus wild-type cells, leading to the conclusion that transcript lengths, rather than m⁶A levels, are correlated with SG enrichment (Khong, *et al.*, *Nat. Commun.*, 2022).”

Moon *et al.* applied single-molecule tracking on reporter mRNAs to uncover co-localization with and bi-directional trafficking between stress granules and processing bodies. They demonstrated that translating mRNAs can transiently co-localize with stress granules, whereas non-translating mRNAs form stable mRNP granule structures. Notably, the *KDM5B* mRNA used in this study was also enriched in our SG CAP-seq datasets in both *HEK293T* and U-2 OS cells. As suggested by the reviewer, we have cited this work along with the Mateju *et al.*, 2020 *Cell* citation in the Introduction. The following comment on *KDM5B* is added to the revised manuscript:

Page 5, lines 13 - 16: “As a demonstration of the high spatial specificity of CAP-seq, known SG-localized transcripts, such as lncRNA *NORAD*, mRNA *DYNC1H1*, and mRNA *KDM5B* (Khong, *et al.*, *Mol. Cell*, 2017; Moon, *et al.*, *Nat. Cell Biol.*, 2019), were enriched in our SG dataset, whereas mitochondrial RNAs were significantly depleted.”

Page 9, lines 1 - 3: “Among 1135 CAP-seq enriched RNAs, 296 (26%) have also been identified as SG transcripts in the previous study, including *AHNAK*, *NORAD*, *PEG3*, *CDK6*, and *KDM5B* (Khong, *et al.*, *Mol. Cell*, 2017; Moon, *et al.*, *Nat. Cell Biol.*, 2019) were also found enrichment in the datasets.”

3. G3BP1 is an RNA binding protein, therefore it is expected that mRNAs that bind to G3BP1 will be present in the unstressed and stressed G3BP1-miniSOG datasets. The statement, “It appears that a sub-population of mRNAs remain proximal to G3BP1 in the cytoplasm despite apparent SG disassembly” implies that these detected RNAs are within stress granule cores, but no experimental evidence is presented to support that statement.

Response: We agree with the reviewer that since G3BP1 is an RNA binding protein, it is expected to interact with mRNAs in both stressed and unstressed states. Indeed, previous live cell proximity-dependent protein labeling experiments have portrayed a pre-existing protein-protein interaction network formed by SG core components in the absence of stress (PMID: 29373831, PMID: 29395067). The concept of SG cores has been further implicated in live cell imaging of the SG disassembly process, which reveals the non-uniform SG disintegration into smaller foci before undergoing further clearance (PMID: 27602576). Given the above literature findings about SG cores, our observation that a sub-population of mRNAs appears to remain proximal to G3BP1 may likely arise from interaction with SG core components, although we do not have experimental data to directly support this speculation. Future super-resolution imaging experiments may provide further evidence for resolving this issue.

In the revised manuscript, we have added the following discussion to clarify the meaning of “proximity to G3BP1”:

Page 22, lines 1 - 5: “Proximity-labeling experiments have also portrayed an extensive protein-protein interaction network formed by SG core components even in the absence of stress (Youn, *et al.*, *Mol. Cell*, 2018; Markmiller, *et al.*, *Cell*, 2018). Since G3BP1 is an RNA binding protein, it is expected to interact with mRNAs in both stressed and unstressed states.”

Page 22, lines 10 - 11: “Live cell imaging of the SG disassembly has revealed the non-uniform SG disintegration into smaller foci before undergoing further clearance (Wheeler, *et al.*, *Elife*, 2016).”

Page 22, lines 22 - 27: “While SGs are typically invisible in most cells after 3 hr of recovery, it remains unknown whether the RNA interaction network has been reset to the basal state. Our observation that a sub-population of mRNAs appears to remain proximal to G3BP1 may likely arise from interactions with SG core components, although we do not have experimental data to directly support this speculation. Future super-resolution imaging experiments may provide further evidence for resolving this issue.”

4. The m6A enrichment data shown in Figure 4H and 5G must be normalized to total transcript length to be meaningful.

Response: We thank the reviewer for the advice. In the revised manuscript, we have re-analyzed our CAP-seq datasets by normalizing the number of m⁶A sites with total transcript length (kb) (i.e. number of sites per kilobase). A similar trend was observed as our previous analysis using m⁶A sites alone, revealing that the density of m⁶A is higher in pre-existing G3BP1-proximal RNAs (Figure S23). During the disassembly stage, the density of m⁶A is also the highest in Cluster 2 (RNAs that dissociate more slowly during the recovery phase), which is consistent with our previous observation using m⁶A sites alone (Figure S26). In the revised manuscript, we have provided these figures in the supplementary file and added the following descriptions in the main text:

Page 14, line 30: “We thus analyzed both the number of m⁶A sites and m⁶A site density (number per kilobase) in our datasets, using published m⁶A database in HEK293T cells.”

Page 17, line 34: “We therefore compared the extent of m⁶A sites and m⁶A site density (number per kilobase) in our dataset.”

Figure S23. Comparing the proportion of mRNA with different m⁶A density (m⁶A sites per kilobase) in basal-specific, pre-existing and *de novo* datasets.

Figure S26. Comparing the proportion of mRNA with different m⁶A density (m⁶A sites per kilobase) between 4 clusters.

5. The details of the CAP-seq approach must be clarified to indicate the limitations of the assay. The following should be addressed to give confidence that the method is resulting in specific labeling of RNAs in stress granules:

- a. The distance between the miniSOG protein and the RNA target that allows labeling should be stated.

Response: We thank the reviewer for pointing out the issue of labeling radius. While the exact radius has not been measured in the context of live cells, the half-life and diffusion radius of singlet oxygen were estimated to be 0.6 μ s and 70 nm, respectively (Moan, *J. Photoch. Photobio.*, 1990). Since the half-life of singlet oxygen depends critically on the local environment and considering the quenching reaction from cellular metabolites such as thiols and amines, the CAP-seq labeling is likely quite restricted. In our previous CAP-seq profiling at the surface of endoplasmic reticulum membrane, 96.2% of the captured mRNAs encode for the secretory pathway proteins, thus

suggesting a spatial resolution on the scale of the size of a ribosome, i.e. 20 nm. As the size of SGs is in the range of 100 nm – 1 μ m, a resolution on the scale of tens of nanometers is sufficient for mapping SG components. In the revised manuscript, we have added the above information of labeling radius in the Introduction:

Page 3, lines 10 - 17: “Due to the limited half-life and diffusion distance of singlet oxygen (0.6 μ s and 70 nm in cells) (Moan, *J. Photoch. Photobio.*, 1990), CAP-seq have been applied in profiling the subcellular transcriptome in the mitochondria and endoplasmic reticulum with high spatial resolution in live cells.”

b. The degree of G3BP1-miniSOG localization to stress granules and the degree of the control miniSOG in stress granules must be quantified and reported, especially considering that miniSOG is present in stress granules.

Response: In the revised manuscript, we performed additional immunofluorescence microscopy experiments to quantify the percentage of G3BP1-miniSOG that co-localizes with the SG marker, G3BP2, in both HEK293T and U-2 OS cells. In cells challenged with arsenite stress, 57 ± 7.5 % and $46\% \pm 2.4\%$ of G3BP1-miniSOG overlaps with SGs in HEK293T and U-2 OS cells, respectively. In contrast, the degree of co-localization with SG is significantly lower for untargeted-miniSOG in these cells (HEK293T: 21 ± 3.2 %; U-2 OS: $9.8\% \pm 0.4\%$). Notably, these values are quite similar to the ratios of RNA enrichment (SG vs. whole cell) in our smFISH experiments (Figure 1F). In the revised manuscript, we have added the above data in Supplementary Figures S2 and S9. We have also added the following information in the main text:

Page 4, lines 7 - 10: “Immunofluorescence imaging confirmed the co-localization of G3BP1-miniSOG (57 ± 7.5 %, mean \pm s.d., $n = 30$ cells) with the SG marker, G3BP2, whereas untargeted miniSOG remains dispersed throughout the cell (21 ± 3.2 %, $n = 30$ cells) (Figure S3).”

Page 7, lines 10 - 15: “Immunofluorescence imaging showed good co-localization between miniSOG and SG marker G3BP2, and $46\% \pm 2.4\%$ ($n = 30$) of the G3BP1-miniSOG fusion protein were localized to SG (Figure S9). In contrast, only $9.8\% \pm 0.4\%$ of untargeted miniSOG protein ($n = 30$) overlaps with SGs (Figure S9). Following RNA labeling, biotinylation signal is highly co-localized with both miniSOG and the SG marker, G3BP2 (Figure 2A).”

c. The definition of ‘post-versus pre-enrichment of RNA’ should be given. Is this equivalent to an ‘input’ sample in an RNA-protein co-immunoprecipitation experiment?

Response: Yes, the “pre-enrichment” sample is the same as “input” sample used in an RNA-protein co-IP experiment. The pre-enrichment RNAs refers to the total RNAs

before loading onto streptavidin-coated beads for affinity purification. The post-enrichment RNAs refer to those after streptavidin beads purification. We have clarified this statement in the manuscript:

Page 5, lines 2 - 4: “(1) post- versus pre-enrichment of RNA labeled with G3BP1-miniSOG (post-enrichment: RNAs eluted from streptavidin-coated beads; pre-enrichment: total RNAs before loading onto streptavidin-coated beads);”

d. The rationale for pooling the three negative controls (post-versus pre-enrichment, PA-omitted, and untargeted miniSOG) should be clarified and results should be shown for each individual treatment as they are controlling for different aspects of the approach.

Response: The rationale of using three negative controls is to provide a stringent list of SG-proximal RNAs that we hope to exclude false positives as much as possible. These controls are necessary to remove non-specific background adsorption to streptavidin-coated beads (by omitting PA as the control) and to remove background RNA labeling outside the SGs (by using untargeted miniSOG as the control). In the revised manuscript, we have provided the results from each control in Figures S4, S7, S11, S15, and S19 (Figure S6, Figure S10, Figure S16, Figure S20 in the latest manuscript), and emphasized the rationale of using these control samples in the main text:

Page 4, lines 30 - 33: “In addition, as G3BP1 was only partially localized to SG under sodium arsenite stress (Wheeler, *et al.*, *Methods*, 2017) (Figure S3), we used HEK293T cells expressing untargeted miniSOG as another control to subtract background RNA labeling in the cytoplasm.”

e. It is unclear whether this same approach of pooling all three negative control datasets is used for the unstressed HEK293T cell and the recovery from stress datasets.

Response: Yes, we applied the same criteria to determine the G3BP1-proximal RNA dataset throughout this study, i.e. for stressed, unstressed, and recovery experiments. For example, Figure 4C shows the Venn diagram of DESeq analysis against all three negative controls. In the revised manuscript, we have clarified this issue by adding the following information to the main text.

Page 13, lines 31 -36: “Following the same DESeq2 work flow and cutoff values ($\log_2FC > 0.3$ and $p_{adj} < 0.05$) as in the previous analysis of stressed cells, CAP-seq identifies 3302, 2136, and 1490 transcripts in post- vs. pre-enrichment, labeling vs. omitting PA, and G3BP1-miniSOG vs. untargeted miniSOG analyses, respectively.

The overlap of these three datasets yields 731 RNAs, which we define as G3BP1-proximal transcripts under the basal condition (Figure 4C, Figure S20, Table S1C).”

Page 17, lines 4 - 9: “Following the identical RNA labeling protocol (10 mM PA and 15 min blue LED illumination), DESeq2 work flow (post- vs. pre-enrichment, labeling vs. omitting PA, and G3BP1-miniSOG vs. untargeted miniSOG) and cutoff values ($\log_2FC > 0.3$ and $p_{adj} < 0.05$) as in the previous analyses of stressed and unstressed cells, a total of 533 and 428 G3BP1-proximal transcripts were captured at T1 and T3, respectively (Figures S24, Table S1D - E)”

6. Poor data quality throughout the manuscript makes the study results difficult to interpret. In particular:

a. Unstressed cells must be shown in Figure 1B, Figure 2A, Figure 3A, Figure S2

Response: We thank the reviewer for this suggestion. We have added fluorescence images of unstressed cells in Figures 1B, 2A, 3A, and S2, as requested by the reviewer.

Figure 1B Immunofluorescence microscopy of HEK293T cells expressing G3BP1-miniSOG and untargeted miniSOG (green). Cells were treated with 0.5 mM sodium arsenite for 60 min and labeled by 10 mM PA under 15 min blue light illumination. Endogenous SG marker G3BP2 and biotinylated signal are shown in cyan and magenta, respectively. Scale bars, 10 μ m.

Figure 2A Immunofluorescence microscopy of U-2 OS cells expressing G3BP1-miniSOG and untargeted miniSOG (green). Cells were treated with 0.5 mM sodium arsenite for 60 min and labeled by 10 mM PA under 15 min blue light illumination. Endogenous SG marker G3BP2 and biotinylated signal are shown in cyan and magenta, respectively. Scale bars, 10 μ m.

Figure 3A Immunofluorescence microscopy of HEK293T cells expressing G3BP1-miniSOG and untargeted miniSOG (green). Cells were treated with 0.4 M sorbitol for 150 min and labeled by 10 mM PA under 15 min blue light illumination. Endogenous SG marker G3BP2 and biotinylated signal are shown in cyan and magenta, respectively. Scale bars, 10 μ m.

Figure S3. Co-localization of G3BP1-miniSOG and untargeted-miniSOG with SG. **(A)** Immunofluorescence microscopy of HEK293T cells expressing G3BP1-miniSOG and untargeted miniSOG (green). Stressed cells were treated with 0.5 mM sodium arsenite for 60 min. Endogenous SG marker G3BP2 or TIA1 signal are shown in cyan. Scale bars, 10 μ m. **(B)** Co-localization of G3BP1-miniSOG and untargeted-miniSOG with the SG marker G3BP2. Quantification was performed with a provided script written in MATLAB. (n = 30 from three biological replicates).

b. Statistical analyses must be done for all Venn diagrams to enable a meaningful interpretation to be made (Figure 1D, Figure 2C, Figure 2D, Figure 3C, Figure 3D, Figure S5, Figure S12, Figure S16; Figure S19, Figure S20)

Response: We thank the reviewer for this suggestion. In the revised manuscript, we applied hypergeometric test to calculate the relevant p-values of Venn diagrams. Figure 2D compares our CAP-seq SG-proximal and SG-excluded RNAs from arsenite-stressed U-2 OS with the previously published SG RNA dataset, the p-values of these comparisons are 2×10^{-183} and 0.0001, respectively. Figure 3D compares CAP-seq datasets for arsenite stress and sorbitol stress. The p-value of this comparison is 2×10^{-150} . Figure S20 compares CAP-seq defined SG-proximal dataset under T0 and G3BP1-proximal datasets under T1 and T3. The p values of comparisons between T0 vs. T1, T0 vs. T3 and T1 vs. T3 are 9×10^{-321} , 1×10^{-150} and 5×10^{-266} , respectively. We also added p value of other Venn diagrams that are not mentioned here. Figure 4D compares the CAP-seq enriched RNAs under unstressed, arsenite stress and sorbitol stress, the p value of the pre-existing datasets is 4×10^{-246} . Figure 6A compares RNAs enriched at basal and T3 time point, the p-value of this comparison is 5×10^{-135} . The p values have been attached to the mentioned Venn diagrams.

We note that the Venn diagrams shown in Figures 1D, 2C, S5, S12, S16, and S19 are used for defining our CAP-seq datasets of G3BP1-proximal or G3BP1-excluded transcripts. In these Venn diagrams, each circle represents a unique control

experiment. The overlap provides a stringent list of RNA with high spatial specificity, rather than representing a comparison of datasets. Thus, we did not calculate the p-values of these Venn diagrams.

c. Loading controls and results from experimental replicates must be included for all western blots (Figure S1, Figure S6, Figure S18)

Response: In the revised manuscript, we have added the loading controls to the Western blots. We performed biological triplicated experiments ($n = 3$) for each blot and reported the mean \pm standard error of image quantitation in the main text. We also added error bars to revised Figure S18 (Figure S2 in the latest supplementary information).

Page 3, lines 33 - 34: “The expression level of G3BP1-miniSOG was $65\% \pm 4.9\%$ (mean \pm s.d.) that of the endogenous G3BP1 level, as measured by western blot (Figure S1)”

Page 7, lines 8 - 9: “Western blot analysis revealed that the expression level of G3BP1-miniSOG was approximately $63\% \pm 2\%$ of the endogenous G3BP1 (Figure S8).”

Figure S1. Western blot detection of G3BP1 overexpression in HEK293T cells stably expressing G3BP1-miniSOG. eIF2α was set as loading control.

Figure S8. Western blot detection of G3BP1 overexpression in U-2 OS cells stably expressing G3BP1-miniSOG. eIF2α was set as loading control.

Figure S2. Western blot analysis of eIF2α phosphorylation (p-eIF2α) in G3BP1-miniSOG HEK293T cells in unstressed (basal), stressed (T0), and recovery (T1 and T3) stages. The intensity of eIF2α phosphorylation are normalized with eIF2α expression. Error bars represent standard deviation.

d. Some smFISH images do not appear to have detectable RNA and/or stress granules in the cytoplasm, making them uninterpretable (Figure 1E *CCNL2*, Figure 2E *SRRM2*, Figure S8 *PKD1*)

Response: Since *CCNL2*, *SRRM2* and *PKD1* are all candidates from our SG-excluded CAP-seq datasets, it is expected that little or no co-localization exists between their smFISH signal and SGs. Figures 1E, 2E, and S8 (S11 in the latest manuscript) are consistent with this expectation. In the revised manuscript, we have emphasized in the figure captions that these transcripts are from SG-excluded datasets.

Figure 1E smFISH of SG-proximal (*GAS1*, *BMS1*, *APLP2*, *USP7*) and SG-excluded mRNAs (*CCNL2*, *PCBP2*). Scale bars, 5 μ m. **(F)** Quantitation of the ratio of smFISH within SG vs. whole cell ($n = 20$ cells from three biological replicates).

Figure 2E smFISH validation of CAP-seq uniquely captured SG-proximal mRNA (*SMC1A*) and SG-excluded mRNA (*SRRM2*). Scale bars, 5 μ m. **(F)** Quantitation of the ratio of smFISH within SG vs. whole cell ($n = 20$ cells from three biological replicates).

Figure S11. smFISH validation of CAP-seq defined SG-excluded mRNAs (*PLXNB2*, *PKD1*, *MT-ND4*) in arsenite-stressed U-2 OS cells. Scale bars, 5 μ m. Quantitation of the ratio of smFISH within SG vs. whole cell ($n = 20$ cells from three biological replicates) is shown on the right.

e. The results of independent experimental replicates for the smFISH data (e.g., Figure 1F) should be reported rather than the number of individual cells quantified.

Response: We thank the reviewer for the advice. The smFISH data in Figure 1F are from three biological replicates with 20 cells in total. In the revised figure caption, we have provided information on the biological replicates:

Figure 1F: “Quantitation of the ratio of smFISH within SG vs. whole cell ($n = 20$ cells from three biological replicates).”

Figure 2F: “Quantitation of the ratio of smFISH within SG vs. whole cell ($n = 20$ cells from three biological replicates).”

Figure 3F: “Quantitation of the ratio of smFISH within SG vs. whole cell ($n = 20$ cells from three biological replicates).”

Figure S11: “smFISH validation of CAP-seq defined SG-excluded mRNAs (*PLXNB2*, *PKD1*, *MT-ND4*) in arsenite-stressed U-2 OS cells. Scale bars, 5 μ m. Quantitation of the ratio of smFISH within SG vs. whole cell ($n = 20$ cells from three biological replicates) is shown on the right.”

Figure S18: “Quantitation of the level of smFISH targets (*MAD2L1*, *PRDX3*, *MORF4L2*) enrichment in SGs ($n = 20$ cells from three biological replicates).”

7. The stress granule transcriptome generated by Khong et al. 2017 Mol Cell is treated throughout the manuscript as if all the RNAs within it are enriched in stress granules, however, this dataset contains all detectable RNAs that are in stress granules at varying degrees. This is an important distinction because CAP-seq results throughout the paper are not generally showing different results from previously published data in Khong et al., 2017.

Response: We agree with the reviewer that, in terms of RNA features that correlate with SG localization, our datasets are overall consistent with previously published data in Khong et al., 2017. The advantage of CAP-seq lies in its high spatiotemporal resolution which enables the detection of G3BP1-proximal transcripts in unstressed state or in the disassembly stage. In the revised manuscript, we have added the following discussion:

Page 19, line 24 - page 20, line 3: “For example, mRNAs *SRRM2*, *PLXNB2*, *PKD1* and *MT-ND4*, were previously enriched from purified SG (Khong, *et al.*, *Mol. Cell*, 2017) but were found to be SG-excluded by CAP-seq. Our quantitative smFISH imaging analysis shows less than 10% co-localization of these RNAs with SG, thus supporting the CAP-seq dataset.”

In the Khong et al., 2017 paper, the SG_{coreRNA} dataset is defined by comparing FPKM values of RNAs from purified SG core against those of the total RNAs. Among the list of 1841 SG_{coreRNA}, however, 8 mitochondrial mRNAs (MT-mRNAs) appear enriched, with Fold Change (SG core vs. total RNA) > 2.5. Notably, *MT-ND5* is ranked

as the top 5 of SG enriched RNAs. These results are surprising as MT-mRNAs are transcribed from the mitochondrial genome and strictly localized within the mitochondrial matrix. Thus, they should be considered as contaminants in the SG_{coreRNA} dataset.

Our CAP-seq datasets were defined by the overlap of three differential analysis, which provides a more stringent list of SG-enriched RNAs. We compared our CAP-seq datasets with the published list, performed FISH imaging of target RNAs *SRRM2*, *PLXNB2*, *PKD1* and *MT-ND4*, which were found to be SG-excluded by CAP-seq but defined by Khong *et al* as SG-enriched. Quantitative analysis of FISH imaging revealed less than 10% co-localization with SG of these RNAs, thus demonstrating the high spatial specificity of CAP-seq.

8. Several statements in the text are misrepresenting the presented data:

The statement “No significant differences in TE and transcript length were found among mRNAs in stress-independent, sorbitol-specific and arsenite-specific datasets” is not supported by the data shown in Figure S14, which shows significant differences in TE and transcript length in the arsenite-specific dataset.

Response: We have double-checked the data in Figure S14 (Figure S19 in the latest manuscript) and confirm that the three datasets (stress-independent, sorbitol-specific, and arsenite-specific) do not show statistical differences in the TE and transcript length (p values > 0.05, Mann-Whitney test). The data used for statistical analysis are listed in Table S3. The statistical differences shown in the Figure are between total mRNAs and some of the above datasets, rather than among the three datasets.

Figure S19. Box plot comparing the translation efficiencies (left) and transcript length (right) in

arsenite-specific, stress-independent, sorbitol-specific SG-proximal mRNAs in HEK293T cells. Translation efficiencies are counted from a previous study (Sidrauski, *et al.*, *Elife*, 2015). AU content and transcript length features are referenced from Ensembl website. ns, not significant ($p > 0.05$); * $p < 0.05$; ** $p < 0.01$; *** $p < 0.001$; **** $p < 0.0001$ (Mann-Whitney test).

9. The statement “At 3 hr post-stress, the eIF2a phosphorylation appears fully restored to its basal level” is not supported by the data in Figure S18 which shows P-eIF2 is 2-3x higher at 3 hr compared to the basal unstressed condition.

Response: We quantified the intensity of eIF2a phosphorylation normalized to the intensity of Tubulin. The level of eIF2a phosphorylation in T3 was $29 \pm 15\%$ higher than basal level. The statement and Figure S2 have been updated in the latest manuscript.

Figure S2. Western blot analysis of eIF2 α phosphorylation (p-eIF2 α) in G3BP1-miniSOG HEK293T cells in unstressed (basal), stressed (T0), and recovery (T1 and T3) stages. The intensity of eIF2 α phosphorylation are normalized with eIF2 α expression. Error bars represent standard deviation.

Minor comments:

1. References to Parker *et al.*, and Jeffrey (Jaffrey) *et al.*, should be replaced with the citation in proper format.

Response: Thanks for pointing out this issue. We have corrected this in the manuscript.

2. The observation that mitochondrial gene encoded RNAs are depleted from stress granules using CAP-seq is not novel as it is stated in Khong *et al.*, *Mol Cell* 2017 that mitochondrial RNAs present in their transcriptome dataset may be contaminants introduced during purification.

Response: In Khong et al, mitochondrial gene encoded RNAs were found to be enriched in the SG, with Fold change (SG core vs. Total RNAs) no less than 2.5.

3. The source of the data used to create translation efficiency and transcript length plots throughout should be defined in the figure legends.

Response: Thanks for pointing out this issue, we have already added this source in the figure legends.

We used same translation efficiency data source as Khong et al. and TRIBE-ID, which counted from Sidrauski, *et al.*, *Elife*, 2015. Transcript length were downloaded from Ensembl website.

Figure 3G: Box plot comparing the length features between sorbitol-specific, stress-independent and arsenite-specific SG-proximal mRNAs. ns, not significant ($p > 0.05$); **** $p < 0.0001$ (Mann-Whitney test). Transcript length features are referenced from Ensembl website.

Figure 4E - G: Bar plot showing the comparison of AU content in different regions (E), length features (F) and translation efficiencies. (G) between mRNAs in basal-specific, pre-existing and *de novo* datasets. ns, not significant ($p > 0.05$); * $p < 0.05$; ** $p < 0.01$; *** $p < 0.001$; **** $p < 0.0001$ (Mann-Whitney test). Translation efficiencies are counted from a previous report (Sidrauski, *et al.*, *Elife*, 2015). Transcript length features and AU content are referenced from Ensembl website.

Same statement is also attached to the legends of Figure 5E - F, Figure 6B - D, Figure S12, Figure S14, Figure S19 and Figure S22.

Reviewer #3 (Remarks to the Author):

In this manuscript, Ren and colleagues use the RNA proximity labeling approach CAP-seq to identify and quantify transcripts localized to stress granules (SG). They use this technique to profile how the SG transcriptome changes in response to different stresses (arsenite vs. sorbitol) and how its RNA content dynamically changes upon SG disassembly. Generally, the conclusions in the manuscript are supported by the data. I have only a few comments that may improve the manuscript.

MAJOR COMMENTS

1. Again and again, the authors find that SG-proximal RNAs are (1) long, (2) AU-rich, and (3) poorly translated. They ascribe function to all of these characteristics and note that they may be involved in localizing RNAs to SG. However, across the transcriptome,

all 3 of these characteristics are correlated with each other. That is, RNAs that are long also tend to be AU-rich, and vice versa (Marin et al Yeast 2003; Lopez et al Frontiers in Genetics 2021, and others). It could be then that just one of these characteristics is important and the others come along for the ride through these correlations but are not actually functional themselves in terms of getting RNAs to SGs. This point should at least be noted.

Response: We thank the reviewer for the valuable advice. Indeed, RNA features such as AU content have been reported to be correlated with transcript length. When analyzing our CAP-seq dataset, in addition to examining the correlation between SG enrichment with total RNA length and AU% at the transcript level, we also break down the transcripts into 5'UTR, CDS, and 3'UTR segments. We observed quite different trends at each segment. For example, stress-de novo RNAs feature both longer length and higher AU% in their 3'UTR, whereas their CDSs are shorter with higher AU% (Figure 4E-F). Another example is the G3BP1-proximal RNA Cluster #4 during SG disassembly, which are characterized with shorter 3'UTR but higher AU%. These observations indicate that AU% and length may contribute differently to SG enrichment, as least in a few cases presented in our study. In the revised manuscript, we have noted this issue to the readers and cited the references of Marin et al Yeast 2003 and Lopez et al Frontiers in Genetics 2021:

Page 22, line 33 - page 23 line 7: "It should be noted that RNA features such as AU content have been reported to be correlated with transcript length (Marin, *et al.*, *Yeast*, 2003). To better examine the correlation between SG enrichment with total RNA length and AU content at the transcript level, we broke down the transcripts into 5'UTR, CDS, and 3'UTR segments. We observed quite different trends at each segment. For example, stress-de novo RNAs feature both longer length and higher AU content in their 3'UTR, whereas their CDSs are shorter with higher AU content (Figure 4E-F). RNAs remained proximal to G3BP1 during final stages of SG disassembly (cluster 4) were also characterized with shorter 3'UTR but more AU. These observations indicate that AU content and length may contribute differently to SG enrichment."

Prompted by the reviewer's advice, we have also sought to disentangle the effects of m⁶A site number and transcript length, which are strongly and positively correlated to each other. In the revised manuscript, we have re-analyzed our CAP-seq datasets by normalizing the number of m⁶A sites with total transcript length (kb) (i.e. number of sites per kilobase). A similar trend was observed as our previous analysis using m⁶A sites alone, revealing that the density of m⁶A is higher in pre-existing G3BP1-proximal RNAs (Figure S23). During the disassembly stage, the density of m⁶A is also the highest in Cluster 2 (RNAs that dissociate more slowly during the recovery phase),

which is consistent with our previous observation using m⁶A sites alone (Figure S26). In the revised manuscript, we have provided these figures in the supplementary file and added the following descriptions in the main text:

Page 14, line 30: “We thus analyzed both the number of m⁶A sites and m⁶A site density (number per kilobase) in our datasets, using published m⁶A database in HEK293T cells.”

Page 17, line 34: “We therefore compared the extent of m⁶A sites and m⁶A site density (number per kilobase) in our dataset.”

Figure S23. Comparing the proportion of mRNA with different m⁶A density (m⁶A sites per kilobase) in basal-specific, pre-existing and *de novo* datasets.

Figure S26. Comparing the proportion of mRNA with different m⁶A density (m⁶A sites per kilobase) between 4 clusters.

2. Related to point 1, I particularly disagree with the statement that “RNA binding proteins generally prefer AU-rich motifs”. Yes, many RBPs do bind AU-rich sequences, but many also do not. Characterizing the entirety of the RNA binding proteome in this way is not accurate.

Response: We thank the reviewer for pointing out this issue. We realize that we should not equate AU% with AU-rich elements, and we agree that it is not accurate to claim that RBPs prefer AU-rich sequences. In the revised manuscript, we have removed the above statement:

Page 14, lines 18 - 22: "In contrast, no significant differences in AU content was observed in the 5' UTR. In terms of transcript length, RNAs in all three G3BP1-/SG-proximal datasets are overall longer than total cellular mRNAs (Figure S22, Table S3B-C), which is in accordance with the previous observation that SG-localized mRNAs are longer than average."

3. Any time that membership in a group is tested across experiments (e.g. Venn diagrams comparing SG-enriched RNAs across stresses or cell types, etc.), p-values should be calculated for the overlap. Without some kind of test, it's difficult for me to know whether or not the overlap observed is more or less than expected. This is important as observing a greater overlap than expected would lend additional confidence to the results.

Response: We thank the reviewer for this suggestion. In the revised manuscript, we have applied hypergeometric test to calculate the relevant p-values of Venn diagrams.

Figure 2D compares our CAP-seq SG-proximal and SG-excluded RNAs from arsenite-stressed U-2 OS with the previously published SG RNA dataset, the p-value of this comparison is 2×10^{-183} and 0.0001, respectively.

Figure 3D compares CAP-seq datasets for arsenite stress and sorbitol stress. The p-value of this comparison is 2×10^{-150} . Figure S20 compares CAP-seq defined SG-proximal dataset under T0 and G3BP1-proximal datasets under T1 and T3. The p values of comparisons between T0 vs. T1, T0 vs. T3 and T1 vs. T3 are 9×10^{-321} , 1×10^{-150} and 5×10^{-266} , respectively.

We also added p value of other Venn diagrams that are not mentioned here. Figure 4D compares the CAP-seq enriched RNAs under unstressed, arsenite stress and sorbitol stress, the p value of the pre-existing datasets is 4×10^{-246} . Figure 6A compares RNAs enriched at basal and T3 time point, the p-value of this comparison is 4×10^{-135} . The p values have been attached to the mentioned Venn diagrams.

MINOR COMMENTS

1. Where did the translational efficiency data come from? I may have missed it, but it was not obvious to me. This should be made clearer.

Response: Thanks for pointing out this issue. We have already added this source in the figure legends.

We used same translation efficiency data source as Khong *et al.* and TRIBE-ID, which counted from Sidrauski, *et al.*, *Elife*, 2015. Transcript length were downloaded from Ensembl website.

Figure 3G: Box plot comparing the length features between sorbitol-specific, stress-independent and arsenite-specific SG-proximal mRNAs. ns, not significant ($p > 0.05$); **** $p < 0.0001$ (Mann-Whitney test). Transcript length features are referenced from Ensembl website.

Figure 4E - G: Bar plot showing the comparison of AU content in different regions (E), length features (F) and translation efficiencies. (G) between mRNAs in basal-specific, pre-existing and *de novo* datasets. ns, not significant ($p > 0.05$); * $p < 0.05$; ** $p < 0.01$; *** $p < 0.001$; **** $p < 0.0001$ (Mann-Whitney test). Translation efficiencies are counted from a previous report (Sidrauski, *et al.*, *Elife*, 2015). Transcript length features and AU content are referenced from Ensembl website.

Same statement is also attached to the legends of Figure 5E - F, Figure 6B - D, Figure S12, Figure S14, Figure S19 and Figure S22.

2. "Transcription factors binding to AU-rich elements [have] been found in SG." Are you sure you mean transcription factors? Those don't normally bind RNA.

Response: Thanks for pointing out this mistake. We have removed this statement in the revised manuscript (page 17, line 21).

3. "However, the purification procedure is prone to contamination and loss of weakly associated material, thus causing high false positive rate." The loss of material that in truth really is associated with SG would result in false negatives, not false positives.

Response: Thanks for pointing out this mistake. We have revised the sentence (page 2, lines 32 - 34): "However, the purification procedure is prone to contamination and loss of weakly associated material, thus causing high false positive and high false negative rates, respectively."

REVIEWERS' COMMENTS

Reviewer #1 (Remarks to the Author):

Throughout the revision period, the authors have carried out additional experiments and made commendable enhancements to the manuscript. Given that the current paper introduces a conventional yet innovative approach for charting spatial RNA molecules, while also pushing the boundaries of our understanding regarding stress granules and the G3BP1 interactome, this reviewer strongly recommends the acceptance of this manuscript in Nature Communications.

Reviewer #2 (Remarks to the Author):

Overall the authors addressed the stated concerns. The remaining issues that must be addressed prior to publication are as follows.

1. While replicates were shown for one of the western blots (Figure S2), the number of independent experimental replicates for the western blots shown in Figure S1, S5, and S8 must be stated.
2. P-eIF2 levels relative to total eIF2 levels from three independent experimental replicates must be presented for Figure S5 to be meaningful.
3. If the unstressed controls now included in Figure 1B, Figure 2A, and Figure 3A were done at a separate time than the other conditions, this must be indicated in the figure legend or it assumed they were done at the same time.
4. Define the term “biological replicates” used throughout the manuscript.
5. The Figure names throughout the text should be checked (for example Figure S9 is mistakenly referenced in Page 9 line 17)

Reviewer #3 (Remarks to the Author):

The authors have addressed my concerns.

REVIEWERS COMMENTS

Reviewer #1 (Remarks to the Author):

Throughout the revision period, the authors have carried out additional experiments and made commendable enhancements to the manuscript. Given that the current paper introduces a conventional yet innovative approach for charting spatial RNA molecules, while also pushing the boundaries of our understanding regarding stress granules and the G3BP1 interactome, this reviewer strongly recommends the acceptance of this manuscript in Nature Communications.

Response: We thank the reviewer for the recommendation.

Reviewer #2 (Remarks to the Author):

Overall the authors addressed the stated concerns. The remaining issues that must be addressed prior to publication are as follows.

1. While replicates were shown for one of the western blots (Figure S2), the number of independent experimental replicates for the western blots shown in Figure S1, S5, and S8 must be stated.

Response: We have added the statement in the figure legend. Three biological replicates were performed for the western blots mentioned above with the raw data provided in the Source Data file.

2. P-eIF2 levels relative to total eIF2 levels from three independent experimental replicates must be presented for Figure S5 to be meaningful.

Response: We have presented the results as request.

Supplementary Figure 5. Western blot analysis of eIF2α phosphorylation (p-eIF2α) in G3BP1-miniSOG HEK293T cells after CAP-seq labeling. The intensities of p-eIF2α are normalized with respect to eIF2α expression. The bars indicate mean values, lines indicate SD. The gel image presented here is a representative example from three independent experiments.

3. If the unstressed controls now included in Figure 1B, Figure 2A, and Figure 3A were done at a separate time than the other conditions, this must be indicated in the figure legend or it assumed they were done at the same time.

Response: We thank the reviewer for pointing out this issue. We have mentioned that unstressed controls were performed at a sperate time than the other conditions in the figure legend.

4. Define the term “biological replicates” used throughout the manuscript.

Response: We thank the reviewer for pointing out this issue. We have changed the statement to independent experiments.

5. The Figure names throughout the text should be checked (for example Figure S9 is mistakenly referenced in Page 9 line 17)

Response: We thank the reviewer for pointing out the mistakes. We have checked the figure names and corrected the mistakes.

Reviewer #3 (Remarks to the Author):

The authors have addressed my concerns.

Response: We thank the reviewer for the comment.